# HIF1A signaling selectively supports proliferation of breast cancer in the brain

Richard Y. Ebright [1,8], Marcus A. Zachariah[1,7,8], Douglas S. Micalizzi [1,2], Ben S. Wittner [1], Kira L. Niederhoffer[1], Linda T. Nieman[1], Brian Chirn[1], Devon F. Wiley[1], Benjamin Wesley[1], Brian Shaw[1], Edwin Nieblas-Bedolla[1], Lian Atlas[1], Annamaria Szabolcs [1], Anthony J. Iafrate[1,3], Mehmet Toner[1,4], David T. Ting [1,2], Priscilla K. Brastianos [1,2], Daniel A. Haber [1,2,5✉] & Shyamala Maheswaran [1,6✉]

Blood-borne metastasis to the brain is a major complication of breast cancer, but cellular pathways that enable cancer cells to selectively grow in the brain microenvironment are poorly understood. We find that cultured circulating tumor cells (CTCs), derived from blood samples of women with advanced breast cancer and directly inoculated into the mouse frontal lobe, exhibit striking differences in proliferative potential in the brain. Derivative cell lines generated by serial intracranial injections acquire selectively increased proliferative competency in the brain, with reduced orthotopic tumor growth. Increased Hypoxia Inducible Factor 1A (HIF1A)-associated signaling correlates with enhanced proliferation in the brain, and shRNA-mediated suppression of HIF1A or drug inhibition of HIF-associated glycolytic pathways selectively impairs brain tumor growth while minimally impacting mammary tumor growth. In clinical specimens, brain metastases have elevated HIF1A protein expression, compared with matched primary breast tumors, and in patients with brain metastases, hypoxic signaling within CTCs predicts decreased overall survival. The selective activation of hypoxic signaling by metastatic breast cancer in the brain may have therapeutic implications.

[1] Massachusetts General Hospital Cancer Center, Harvard Medical School, Charlestown, MA 02129, USA. [2] Department of Medicine, Massachusetts General Hospital, Harvard Medical School, Boston, MA 02114, USA. [3] Department of Pathology, Massachusetts General Hospital, Harvard Medical School, Boston, MA 02114, USA. [4] Center for Bioengineering in Medicine, Massachusetts General Hospital and Harvard Medical School, and Shriners Hospital for Children, Boston, MA 02114, USA. [5] Howard Hughes Medical Institute, Chevy Chase, MD 20815, USA. [6] Department of Surgery, Massachusetts General Hospital, Harvard Medical School, Boston, MA 02114, USA. [7] Present address: Department of Neurosurgery, University of Mississippi Medical Center, Jackson, MS 39216, USA. [8] These authors contributed equally: Richard Y. Ebright, Marcus A. Zachariah. ✉email: dhaber@mgh.harvard.edu; maheswaran@helix.mgh.harvard.edu

Brain metastases occur in about ten percent of all patients with cancer and in as many as a third of women with advanced metastatic breast cancer[1]. As new therapeutic advances increasingly succeed in suppressing cancer progression systemically, recurrence of disease within the central nervous system is emerging as a major cause of cancer relapse and mortality. Systemically administered cancer therapies often lack efficacy within the brain, a phenomenon that may be attributable, in some cases, to poor drug penetration across the blood–brain barrier (BBB), as well as to the acquisition of new mutations in brain metastases that result in distinct drug susceptibility patterns[2]. However, the contribution of the brain microenvironment to metastatic growth and drug resistance is not well defined.

The metastatic cascade begins with cancer cell migration and invasion from the site of the primary tumor into the bloodstream, survival within the high stress circulatory environment, extravasation from blood capillaries into distant organs, and finally adaptation to the unique microenvironment of the metastatic site[3]. In breast cancer, a number of studies have focused on tropism of circulating cancer cells to the brain, including molecular mechanisms that may enable their invasion through the BBB[4–6]. However, organ-specific metastasis may also reflect unique microenvironmental properties and tissue-specific proliferative pathways that contribute to the differential ability of widely disseminated cancer cells to proliferate in some organs while remaining dormant in others[7–10]. Compared with other tissues that are common sites of breast cancer metastasis, such as bone, lung, and liver, the normal brain has relatively low oxygen tension, high glucose-based metabolism, and low collagen content[11,12], factors that may affect the proliferation of cancer cells that have disseminated to the brain.

The cellular response to low oxygen tension is driven by the hypoxia-inducible transcription factors HIF1A and HIF2A, which promote survival, metabolic reprogramming, and angiogenesis in hypoxic environments[13–15]. Hypoxic signaling in primary tumors promotes tumor cell dissemination from the primary tumor—a well characterized role of HIF-mediated epithelial–mesenchymal transition (EMT) on cell migration—and invasion[16,17]. However, the role of hypoxic signaling in later stages, including metastasis initiation, is not as well understood. In brain metastasis models[18], VEGF and other angiogenic growth factors contribute to the development of brain metastases[19,20], but the role of HIF signaling in mediating tumor growth remains uncertain[21].

Circulating tumor cells (CTCs) constitute the metastatic precursors for the blood-borne spread of breast cancer to the brain. These cells are relatively rare in the circulation, but they can be isolated with preserved viability using microfluidic technologies[22]. We previously reported the characterization of cultured breast CTC-derived cell lines, with preserved patient-specific genetic composition and high tumorigenicity in immunosuppressed mice[23–25]. Here, we used serial tumor enrichment of breast CTC lines to generate derivatives with proficiency for proliferation in the brain, and we identify hypoxia and HIF1A pathways as selectively upregulated. We corroborate these findings in clinical brain metastasis samples, in which we observe increased HIF1A and hypoxic signaling versus matched primary breast tumor samples. Furthermore, in patients with brain metastases, increased hypoxic signaling within CTCs is correlated with decreased overall survival. Suppression of HIF1A signaling in breast CTCs abrogates their tumorigenesis in the brain without affecting orthotopic proliferation in the mammary gland, revealing a differential requirement for hypoxic signaling in the brain environment.

## Results
**Breast CTC cultures exhibit differential proliferation rates following intracranial inoculation.** To directly test the proliferative

properties of breast cancer-derived CTCs in the brain, we established a model for stereotactic injection of GFP- and luciferase-tagged cells into the right frontal lobe of immunosuppressed NSG mice. Seven different CTC lines, cultured from the peripheral blood of women with hormone receptor-positive (HR+) metastatic breast cancer (Supplementary Data 1), were each injected into mouse brains (Fig. 1a)[23,24]. Brain tumors became visible by in vivo luciferase imaging at various intervals, ranging from 1 to >10 weeks, with dramatic differences in the rate of growth among the seven breast CTC lines tested. The growth of CTC-derived tumors in the brain was not correlated with their respective proliferation rates in vitro (Fig. 1b). Two CTC lines (Brx-29, Brx-42) demonstrated rapid growth requiring euthanasia of the mice within 6 weeks following intracranial injection; interestingly, both of these lines were derived from blood samples of women who had intracranial metastases at the time of the blood draw. Moderate growth was demonstrated by two other CTC lines (Brx-50, Brx-82), one of which was derived from a patient with multiple intracranial metastases (Brx-82). The three remaining CTC lines (Brx-7, Brx-68, Brx-142) demonstrated slow growth, and none were derived from patients with brain metastases. Histological analysis of CTC-derived brain tumors shows features commonly seen in human brain metastases from breast cancer, including sharp demarcation between tumor and normal brain parenchyma and tumor cell morphology similar to that seen in primary breast tumors (Fig. 1c, d)[26]. Moderate-growth Brx-82 tumors had significantly increased levels of the proliferation marker Ki-67 versus slow-growth Brx-142 tumors ($P = 0.012$) but unchanged levels of the apoptosis marker cleaved caspase-3 ($P = 0.893$), suggesting a proliferative advantage in the brain, with no change in apoptotic index (Fig. 1e, f).

To identify mechanisms that promote the proliferation of breast cancer patient-derived CTCs in the brain within the context of isogenic backgrounds, we undertook serial injections of individual CTC lines with either moderate (Brx-50, Brx-82) or slow (Brx-142) intracranial proliferation, deriving F1 and F2 progeny with increased competence to grow rapidly in the brain. Among the slow-growth CTC lines, Brx-142 was selected based on its rapid in vitro growth. To generate these brain-proficient CTCs, initial CTC-derived brain tumors were harvested and sorted for GFP expression to remove mouse cells, expanded in vitro under anchorage-independent conditions for less than 4 weeks as F1 lines, and then reinjected intracranially to generate F1 brain tumors; this process was then repeated to generate F2 lines and F2 brain tumors (Fig. 2a). Compared to their respective parental CTCs, Brx-82 F1 and F2 lines grow more rapidly in the brain and demonstrate increased lethality (Fig. 2b). Notably, these F1 and F2 lines do not demonstrate increased proliferation in vitro or increased tumor growth orthotopically in the mammary gland; to the contrary, brain-proficient F1 and F2 derivatives grow more slowly in vitro and form slower-growing tumors in the mammary gland compared to parental cells (Fig. 2c, d). This pattern of increased brain-specific F1 growth, with unchanged or reduced mammary and in vitro growth, was observed across all three independent CTC lines for which F1 lines were derived (Fig. 2e, f and Supplementary Fig. 1).

**Hypoxia-associated signaling is upregulated in brain-proficient breast cancer cells.** To characterize pathways that contribute to enhanced proliferation in the brain, we first undertook cancer gene sequencing of the parental, F1 and F2 CTC lines, analyzing 104 oncogenes and tumor suppressors for single-nucleotide variants, insertion–deletion mutations, and copy-number variations (Supplementary Data 2)[27]. No new mutations were detected in the Brx-50 or Brx-142 F1 lines compared with their respective parental lines. In the Brx-82 F1 line, only one acquired mutation

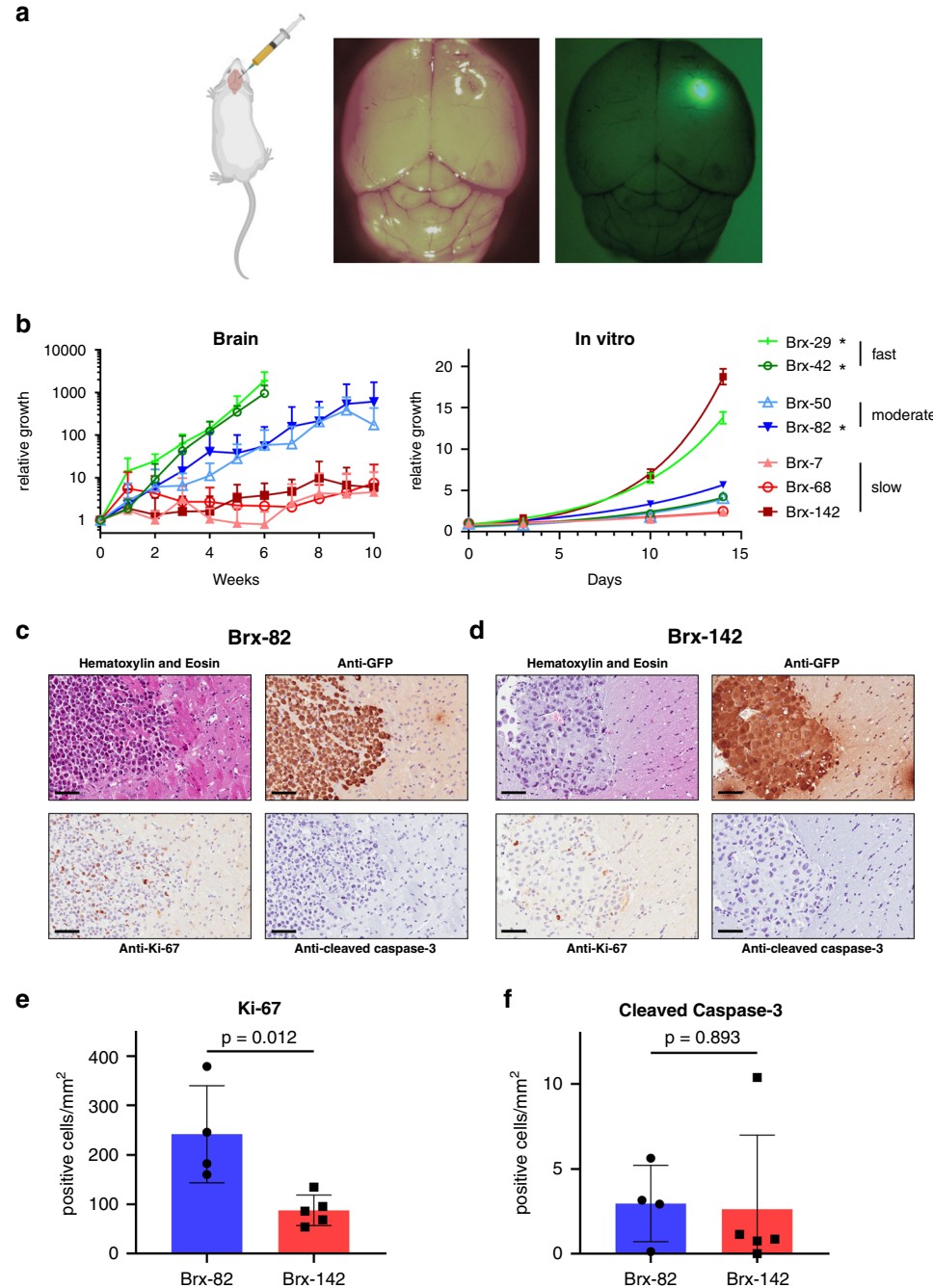

**Fig. 1 Breast cancer CTC lines generate tumors after stereotactic brain injection. a** CTC lines were stereotactically injected into mouse right frontal lobes. CTCs were labeled with GFP and luciferase, allowing for tumor growth monitoring via in vivo imaging and for tumor identification ex vivo. **b** Growth rates of CTCs after stereotactic injection into mouse brains or in vitro. CTC lines were categorized as fast, moderate, or slow growth based on their brain growth rates (in vivo: Brx-7: $n = 5$, Brx-29: $n = 3$, Brx-42: $n = 8$, Brx-50: $n = 7$, Brx-68: $n = 8$, Brx-82: $n = 4$, Brx-142: $n = 6$; in vitro: $n = 5$). Brx-29 and Brx-42 brain growth data were censored at 6 weeks due to rapid tumor growth requiring euthanasia. *CTC lines derived from patients with brain metastases.
**c**, **d** Representative sections of Brx-82 (**c**) or Brx-142 (**d**) brain tumor histology after staining with hematoxylin and eosin; or with anti-GFP, anti-Ki-67, or anti-cleaved caspase 3 antibody (brown) and counter-stained with hematoxylin. Scale bars: 70 μm. Images are representative of four tumor samples.
**e**, **f** Quantitation of the number of cells positive for Ki-67 (**e**) or cleaved caspase-3 (**f**) per mm² by immunohistochemical staining of brain tumor histologic sections (Brx-82: $n = 4$; Brx-142: $n = 5$). P values calculated by two-tailed unpaired $t$-test. Data for in vitro experiments represent mean ± SD and for in vivo experiments represent mean ± SEM. Source data are provided as a Source Data file.

was detected (ALK Gln39Pro; mutant allele frequency 0.518), but it was subsequently lost in the F2 line, and it has not previously been identified as an *ALK* driver mutation, suggesting that it is not linked to the brain proliferative phenotype[28]. These findings, demonstrating absence of mutations in known oncogenes or tumor suppressors, suggest that increased competence of the F1

and F2 derivative lines to grow in the brain may involve non-genetic mechanisms.

We next used RNA-sequencing (RNA-seq) of Brx-50 F1 and Brx-82 F1 versus parental lines to identify cellular signaling pathways correlated with the brain-proficient phenotype (Brx-50: 185 genes up in F1, 130 genes up in parental; Brx-82: 161 genes

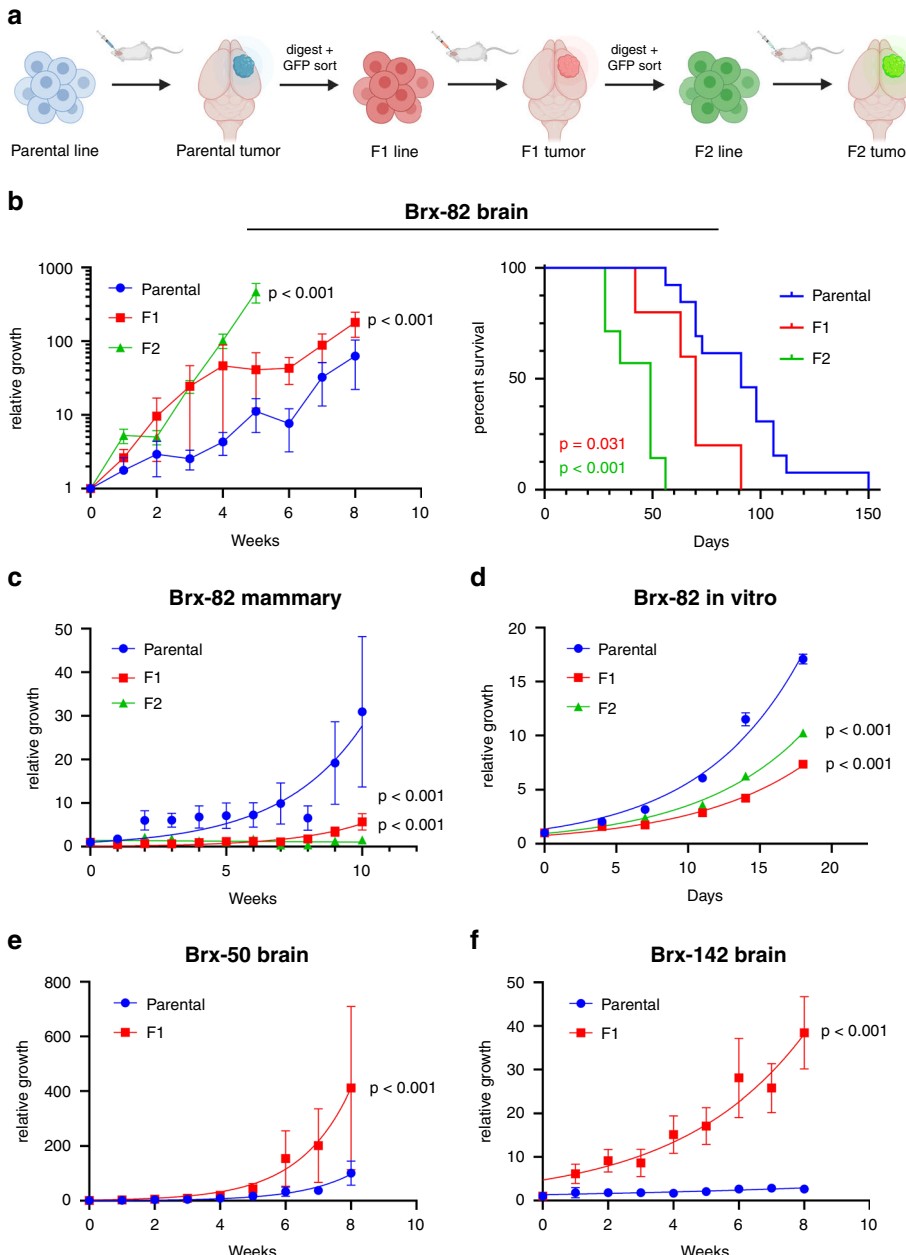

**Fig. 2 CTC lines generated from serial injections into the brain exhibit preferential brain growth. a** Diagram demonstrating the generation of F1 and F2 CTC lines and tumors. **b** Left panel: Brain luminescence monitoring of NSG mice following stereotactic brain injection of Brx-82 parental, F1 or F2 cells (parental: $n = 7$, F1: $n = 5$, F2: $n = 3$). $P$ values calculated by the extra sum-of-squares F test. Right panel: Kaplan–Meier analysis of the survival of mice following stereotactic brain injection of Brx-82 parental, F1 or F2 cells (parental: $n = 13$, F1: $n = 5$, F2: $n = 7$). $P$ values calculated by log rank test. $P$ values indicate comparisons of parental and F1 and of parental and F2. **c** Mammary luminescence monitoring of NSG mice following orthotopic injection of Brx-82 parental, F1 or F2 cells (parental: $n = 20$, F1: $n = 12$, F2: $n = 14$). $P$ values calculated by the extra sum-of-squares F test. $P$ values indicate comparisons of parental and F1 and of parental and F2. **d** In vitro growth of Brx-82 parental, F1 or F2 cells ($n = 5$). $P$ values calculated by the extra sum-of-squares F test. $P$ values indicate comparisons of parental and F1 and of parental and F2. **e** Brain luminescence monitoring of NSG mice following stereotactic brain injection of Brx-50 parental or F1 cells (parental: $n = 10$, F1: $n = 7$). $P$ value calculated by the extra sum-of-squares F test. **f** Brain luminescence monitoring of NSG mice following stereotactic brain injection of Brx-142 parental or F1 cells (parental: $n = 9$, F1: $n = 6$). $P$ value calculated by the extra sum-of-squares F test. Data represent mean ± SEM. Source data are provided as a Source Data file.

up in F1, 363 genes up in parental; fold change >2, FDR < 0.25). Gene set enrichment analysis (GSEA) for pathways enriched within the Molecular Signatures Database hallmarks of cancer genesets[29] identifies three genesets enriched in both Brx-50 F1 and Brx-82 F1 lines: Hypoxia, KRAS Signaling Up, and TNFα Signaling via NF-κB (FDR < 0.25). Of these, the Hypoxia pathway is the most enriched pathway across both F1 lines (Table 1 and Fig. 3a). For both Brx-50 and Brx-82, the vast majority of genes in

the Hypoxia pathway are upregulated in the F1 lines, and Hypoxia genes are among the most enriched genes in both F1 lines (Fig. 3b, c). Furthermore, the Brx-50 F1 line also demonstrates enrichment of the angiogenesis and glycolysis pathways, both of which are known to be regulated by hypoxic signaling (Table 1)[13].

Given the enrichment of hypoxic signaling in brain-proficient F1 lines, we analyzed RNA-seq data from the seven parental CTC

**Table 1 F1 cells demonstrate increased hypoxic signaling: GSEA of RNA-seq from Brx-50 and Brx-82, parental and F1 cells for pathways enriched within the Molecular Signatures Database hallmarks of cancer genesets. NES: normalized enrichment score. NOM P-val: nominal P value.**

| Hallmark genesets | Brx-50 NES | Brx-50 NOM P-val | Brx-50 FDR q-val | Brx-82 NES | Brx-82 NOM P-val | Brx-82 FDR q-val |
|---|---|---|---|---|---|---|
| Enriched in F1 cells | | | | | | |
| HALLMARK_HYPOXIA | 1.916 | <0.001 | <0.001 | 1.718 | <0.001 | 0.019 |
| HALLMARK_ANGIOGENESIS | 1.654 | <0.001 | 0.009 | 1.198 | 0.201 | >0.25 |
| HALLMARK_HEDGEHOG_SIGNALING | 1.526 | 0.002 | 0.033 | −0.727 | >0.25 | >0.25 |
| HALLMARK_TNFA_SIGNALING_VIA_NFKB | 1.551 | 0.004 | 0.033 | 1.324 | 0.036 | 0.239 |
| HALLMARK_CHOLESTEROL_HOMEOSTASIS | 1.438 | 0.009 | 0.070 | −0.725 | >0.25 | >0.25 |
| HALLMARK_IL2_STAT5_SIGNALING | 1.447 | 0.005 | 0.060 | 1.175 | 0.140 | >0.25 |
| HALLMARK_MYOGENESIS | 1.417 | 0.007 | 0.064 | 0.930 | >0.25 | >0.25 |
| HALLMARK_INFLAMMATORY_RESPONSE | 1.456 | 0.020 | 0.066 | 1.020 | >0.25 | >0.25 |
| HALLMARK_KRAS_SIGNALING_UP | 1.372 | 0.026 | 0.095 | 1.644 | <0.001 | 0.037 |
| HALLMARK_GLYCOLYSIS | 1.342 | 0.012 | 0.112 | −0.940 | >0.25 | >0.25 |
| HALLMARK_ALLOGRAFT_REJECTION | 1.331 | 0.024 | 0.113 | 1.330 | 0.050 | >0.25 |
| HALLMARK_P53_PATHWAY | 1.255 | 0.069 | 0.198 | 0.924 | >0.25 | >0.25 |
| HALLMARK_WNT_BETA_CATENIN_SIGNALING | 1.230 | 0.159 | 0.210 | 1.185 | 0.193 | >0.25 |
| HALLMARK_APICAL_JUNCTION | 1.238 | 0.094 | 0.211 | 0.889 | >0.25 | >0.25 |
| HALLMARK_PANCREAS_BETA_CELLS | 1.203 | 0.177 | 0.237 | −1.176 | >0.25 | >0.25 |
| HALLMARK_COMPLEMENT | −0.966 | >0.25 | >0.25 | 1.403 | 0.023 | 0.194 |
| HALLMARK_COAGULATION | 1.110 | >0.25 | >0.25 | 1.319 | 0.070 | 0.211 |
| HALLMARK_PROTEIN_SECRETION | −0.741 | >0.25 | >0.25 | 1.425 | 0.018 | 0.215 |
| Enriched in parental cells | | | | | | |
| HALLMARK_ESTROGEN_RESPONSE_LATE | −1.423 | 0.032 | 0.129 | −1.121 | 0.244 | >0.25 |
| HALLMARK_UV_RESPONSE_DN | −1.449 | 0.016 | 0.130 | 1.260 | >0.25 | >0.25 |
| HALLMARK_SPERMATOGENESIS | −1.461 | 0.025 | 0.151 | −1.748 | <0.001 | <0.001 |
| HALLMARK_XENOBIOTIC_METABOLISM | −1.335 | 0.063 | 0.164 | 1.149 | >0.25 | >0.25 |
| HALLMARK_MYC_TARGETS_V1 | −1.351 | 0.044 | 0.167 | −1.176 | 0.146 | >0.25 |
| HALLMARK_G2M_CHECKPOINT | −1.480 | 0.031 | 0.174 | −1.742 | <0.001 | <0.001 |
| HALLMARK_E2F_TARGETS | −1.588 | 0.012 | 0.222 | −1.780 | <0.001 | <0.001 |
| HALLMARK_ESTROGEN_RESPONSE_EARLY | −1.494 | 0.019 | 0.243 | −0.762 | >0.25 | >0.25 |
| HALLMARK_OXIDATIVE_PHOSPHORYLATION | −1.248 | 0.097 | 0.244 | −1.118 | >0.25 | >0.25 |
| HALLMARK_INTERFERON_ALPHA_RESPONSE | 0.990 | >0.25 | >0.25 | −1.604 | <0.001 | 0.002 |
| HALLMARK_INTERFERON_GAMMA_RESPONSE | 1.072 | >0.25 | >0.25 | −1.481 | 0.003 | 0.039 |
| HALLMARK_MITOTIC_SPINDLE | −0.938 | >0.25 | >0.25 | −1.391 | 0.013 | 0.116 |

lines to determine whether increasing levels of hypoxic signaling correlates with their differential growth rates in the brain microenvironment. Remarkably, the two fast-growth lines display significantly higher hypoxic signaling compared to the three slow-growth lines ($P = 0.015$); the fast-growth lines show a similar trend towards increased hypoxic signaling compared to the two moderate-growth lines ($P = 0.070$) (Fig. 3d). Furthermore, the fast-growth lines also display increased angiogenic ($P = 0.014$) and glycolytic ($P = 0.050$) signaling versus slow-growth lines (Fig. 3d). Thus, activation of hypoxic signaling and its downstream pathways not only are observed following serial brain-enrichment in isogenic CTCs, but also serve as distinguishing features for brain proliferation in multiple independent CTC lines directly enriched from patient blood samples.

Hypoxic signaling regulates cellular metabolic activity, with a shift towards glycolysis and reduced oxygen consumption[13]. We quantified changes in metabolic activity in the derivative lines, observing an increase in the lactate-to-pyruvate ratio in F1 lines, indicating elevated glycolytic activity in these CTC lines compared to parental cells (Fig. 3e and Supplementary Fig. 2). Similarly, using the Seahorse XF platform to measure the rate of oxygen consumption by live cells and quantify active oxidative phosphorylation, we found that oxygen consumption is reduced in F1 lines, consistent with increased hypoxic signaling in these cells (Fig. 3f and Supplementary Fig. 3). These observations demonstrate that increased expression of hypoxic signaling pathways in cultured brain-proficient CTCs is associated with

the expected metabolic shifts toward increased glycolysis and reduced oxygen consumption.

To extend these observations to established models of breast cancer metastasis to the brain, we analyzed previously published transcriptomic data (GSE12237) generated following intracardiac murine injections of the triple negative breast cancer (TNBC) cell line MDA-MB-231 and the HR+ breast cancer cell line CN34[4]. Consistent with our findings in CTC cultures, GSEA of genes differentially upregulated in brain-tropic MDA-MB-231 and CN34 cells compared with parental cells identifies pathways involved in hypoxic signaling ($P = 6.24 \times 10^{-10}$), angiogenesis ($P = 4.13 \times 10^{-4}$) and glycolysis ($P = 1.33 \times 10^{-3}$) (Supplementary Fig. 4). The enrichment of these pathways in a brain-tropic TNBC cell line suggests increased hypoxic signaling in brain metastasis may not be restricted to HR+ breast cancer subtypes.

**HIF1A expression is elevated in brain metastases from breast cancer.** The transcription factor HIF1A is a canonical master-regulator of hypoxic signaling, translocating to the nucleus under hypoxic conditions and regulating multiple pathways, including angiogenesis and glycolysis[13]. Based on our results from CTC lines cultured in vitro, we stained histological sections of CTC-derived tumors for HIF1A expression. Immunohistochemical analysis of both Brx-82 and Brx-142 CTC-derived tumors growing in the brain and the mammary gland reveals increased nuclear HIF1A staining within brain tumors versus mammary

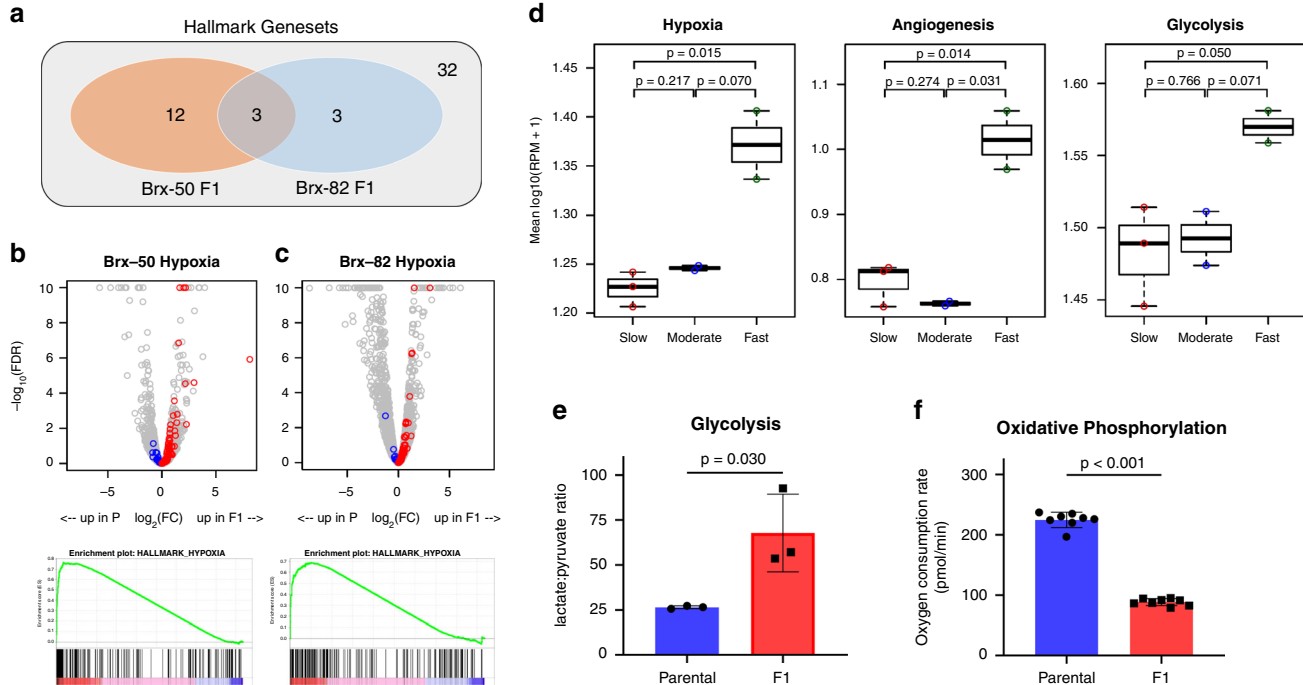

**Fig. 3 Brain-proficient CTC lines demonstrate increased hypoxic signaling. a** Venn diagram of Gene Set Enrichment Analysis (GSEA) of transcripts upregulated in Brx-50 or Brx-82 F1 versus parental cells. Enriched genesets within the Hallmarks of Cancer gene sets from the Broad Molecular Signatures Database are highlighted. **b, c** Top panels: Volcano plots for expression of genes in Brx-50 (**b**) or Brx-82 (**c**) parental and F1 cells, as determined by RNA-seq. Genes in the Hallmarks of Cancer Hypoxia geneset from the Broad Molecular Signatures Database are colored, with blue indicating higher expression in parental cells and red indicating higher expression in F1 cells. Gray markers represent all genes not in the Hypoxia geneset. Genes with -log$_{10}$(FDR) > 10 are displayed as −log$_{10}$(FDR) = 10. Bottom panels: Enrichment plots of the Hallmarks of Cancer Hypoxia geneset for genes enriched in Brx-50 (**b**) or Brx-82 (**c**) F1 versus parental cells. Positive enrichment scores in enrichment plots indicate more enrichment of the geneset in F1 cells. **d** Mean expression of genes in the Hallmark of Cancer Hypoxia, angiogenesis, and glycolysis genesets in the seven parental breast cancer CTC lines, as determined by RNA-seq (Slow: Brx-7, Brx-68, Brx-142; Moderate: Brx-50, Brx-82; Fast: Brx-29, Brx-42). Boxplots display median, 25th percentile, and 75th percentile, with whiskers representing minimum and maximum. *P* values calculated by two-tailed unpaired *t*-test. **e** Lactate-to-pyruvate ratio in Brx-82 parental and F1 cells, as determined by metabolomic studies for relative levels of polar metabolites (*n* = 3). *P* value calculated by two-tailed unpaired *t*-test. **f** Oxygen consumption rate of Brx-82 parental and F1 cells, as determined by live-cell Seahorse assays (*n* = 8). *P* value (1.19 × 10$^{-13}$) calculated by two-tailed unpaired *t*-test. Data represent mean ± SD. Source data are provided as a Source Data file.

tumors (average brain: 46% HIF1A+; average mammary: 4.0% HIF1A+) (Fig. 4a, b and Supplementary Figs. 5 and 6). Transcripts encoding canonical HIF1A target genes are also increased in brain tumors versus mammary tumors. These include genes associated with general hypoxic response (e.g., EGLN3, CA9), glycolysis (e.g., ALDOC, PGK2), and angiogenesis (e.g., TGFB3, VEGFA) (Fig. 4c). Furthermore, GSEA of tumor RNA-seq identifies genes with HIF1A transcription factor binding sites as enriched within brain tumors versus mammary tumors, using two different genesets for HIF1A target genes from the Molecular Signatures Database Transcription Factor Targets database (HIF1_Q3: *P* = 0.002; HIF1_Q5: *P* = 0.038) (Supplementary Fig. 7 and Supplementary Data 3). Notably, RNA levels of HIF1A are unchanged, suggesting that HIF1A protein expression in these brain tumors is primarily regulated at the post-transcriptional level, consistent with its known activation pattern (Fig. 4c)[13].

To extend our results from mouse models to clinical samples, we obtained matched primary breast cancer and brain metastasis samples from six patients with HR+ and/or HER2-amplified breast cancer (average 6.2 years between primary diagnosis and brain metastasis, range: 2.1–15.2 years) (Supplementary Data 4). We performed immunohistochemical analyses for HIF1A protein expression on 1000 cells per sample, scanning for nuclear HIF1A expression using automated imaging to normalize and quantify signal intensity (Fig. 4d). Across the six matched samples, HIF1A nuclear staining intensity was significantly increased in the brain

metastases in five samples (average brain: 69.7% HIF1A+; average mammary: 47.5% HIF1A+) (Fig. 4e and Supplementary Fig. 8). Increased nuclear HIF1A staining in matched brain versus breast tumors also shows a trend toward significant correlation with the number of brain metastases detected in each patient (*r* = 0.799; *P* = 0.057) (Supplementary Fig. 9). To extend these findings to larger clinical databases, we interrogated gene expression profiling data (GSE100534) available from primary patient breast tumors and unmatched brain metastases[30]. Canonical HIF1A target genes are significantly increased in brain metastases compared with primary breast tumors (Fig. 4f). These HIF1A downstream targets include genes associated with general hypoxic response (e.g., CA9, TGM2), glycolysis (e.g., GPI, PGK1), and angiogenesis (e.g., ANGPTL4, VEGFA). Furthermore, as in the analysis of our mouse model, GSEA of these human tumor-derived transcriptomic data identifies genes with HIF1A transcription factor binding sites as enriched within brain metastases versus primary breast tumors (HIF1_Q3: *P* = 0.055; HIF1_Q5: *P* = 0.018) (Supplementary Fig. 10 and Supplementary Data 5). Together, these findings indicate that HIF1A and its downstream effectors are increased in human breast cancer brain metastases compared with primary breast tumors. In addition to enrichment of HIF1A signaling, several other signaling pathways (SREBP1, E2F, AP2, NFY, and CDPCR3) also demonstrate enrichment in brain tumor samples across both our mouse and patient analyses (Supplementary Data 3 and 5).

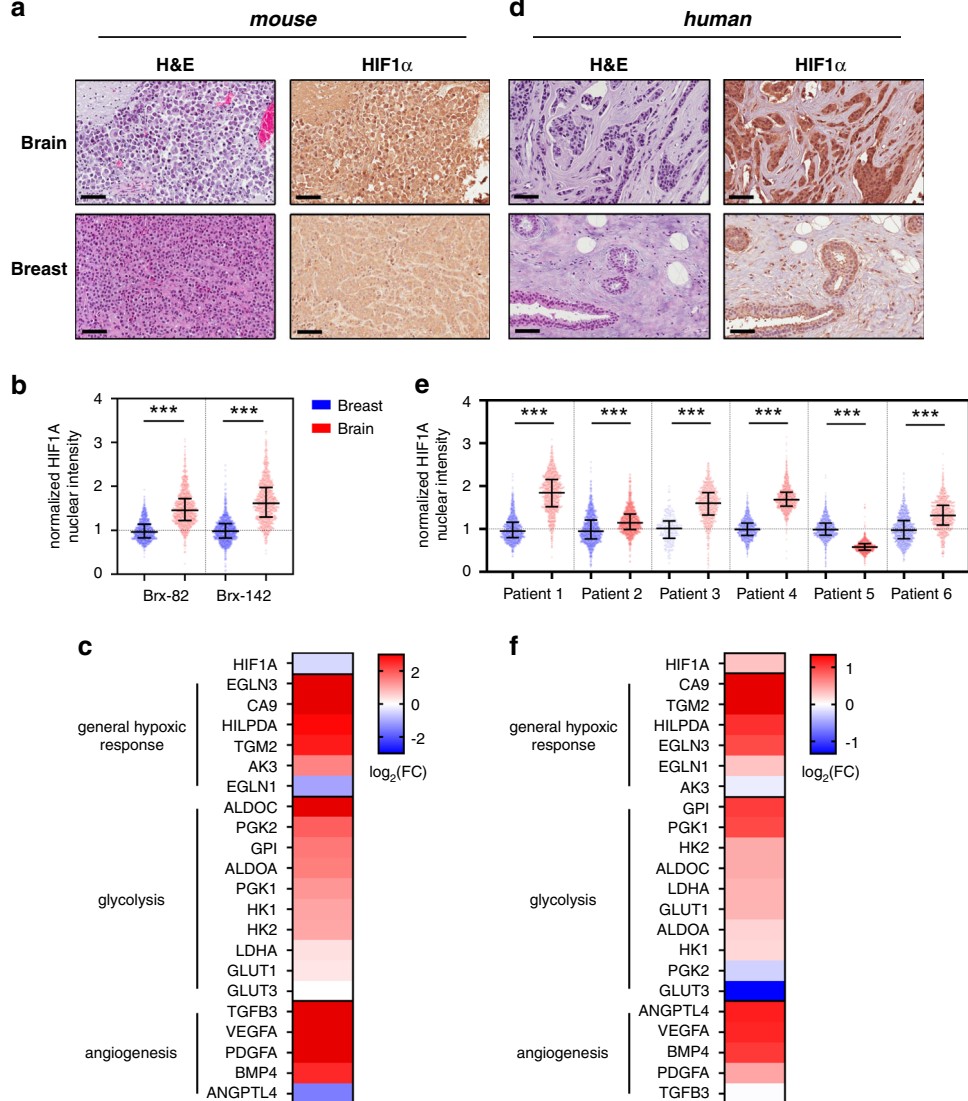

**Fig. 4 HIF1A levels and downstream effectors are increased in brain metastases versus primary breast tumors. a** Representative sections showing Brx-82 brain or breast tumor histology after staining with hematoxylin and eosin; or with anti-HIF1A antibody (brown) and counter-stained with hematoxylin. Scale bars: 70 µm. Images are representative of four tumor samples. **b** Quantitation of nuclear HIF1A staining in Brx-82 and Brx-142 brain and breast tumors, as determined by automated immunohistochemical staining quantitation ($n = 1000$). Median normalized staining with interquartile range displayed. $P$ values (Brx-82: $5.21 \times 10^{-206}$; Brx-142: $1.28 \times 10^{-242}$) calculated by two-tailed unpaired $t$-test. **c** Heat map representing the fold change of Brx-82 brain tumors relative to breast tumors for select HIF1A downstream effectors involved in general hypoxic response, glycolysis, and angiogenesis. **d** Representative sections of matched patient brain metastasis and primary breast tumor histology after staining with hematoxylin and eosin; or with anti-HIF1A antibody (brown) and counter-stained with hematoxylin. Scale bars: 70 µm. Samples displayed are from Patient 4. Images are representative of one tumor sample. **e** Quantitation of nuclear HIF1A staining in matched patient brain metastases and primary breast tumors, as determined by automated immunohistochemical staining quantitation (Patient no. 3 breast: $n = 262$; all other samples: $n = 1000$). Bars represent median, 25th percentile, and 75th percentile normalized staining. $P$ values (Patient 1: $<1 \times 10^{-300}$; Patient 2: $1.39 \times 10^{-36}$; Patient 3: $8.49 \times 10^{-87}$; Patient 4: $2.89 \times 10^{-23}$; Patient 5: $<1 \times 10^{-300}$; Patient 6: $5.08 \times 10^{-93}$) calculated by two-tailed unpaired $t$-test. **f** Heat map representing the fold change of select HIF1A downstream effectors for general hypoxic response, glycolysis, and angiogenesis in patient brain metastases relative to unmatched primary breast tumors[30]. ***$P < 0.001$. Source data are provided as a Source Data file.

**HIF1A is required for preferential growth of breast cancer cells in the brain**. To assess the relative contributions of HIF1A to growth in the brain microenvironment versus in the mammary gland, we sought to determine the relative effects of HIF1A loss on tumor growth at either site. We infected CTCs with lentiviral shRNA constructs targeting HIF1A (shHIF1A) or scrambled control (shCtrl), combining the two lines in a 1:1 ratio, followed by injection of the mixture into either brain or mammary gland. Tumor growth was monitored using in vivo imaging, and tumors were harvested when they had grown to 100 times the original

injection bioluminescent signal. The ratio of shHIF1A-to-shCtrl cells in the output was determined by next-generation sequencing, using the hairpin sequences as barcodes to identify cells harboring either shHIF1A or shCtrl hairpins (Fig. 5a). Two different shHIF1A hairpins were used in these experiments, both of which led to *HIF1A* levels reduced to less than 25% of that of shCtrl lines (Supplementary Fig. 11). GSEA of transcripts differentially expressed between shHIF1A and shCtrl cells (fold change >2; FDR < 0.25) again validates *HIF1A* knockdown and shows decreased hypoxia-associated signaling, as well as decreased

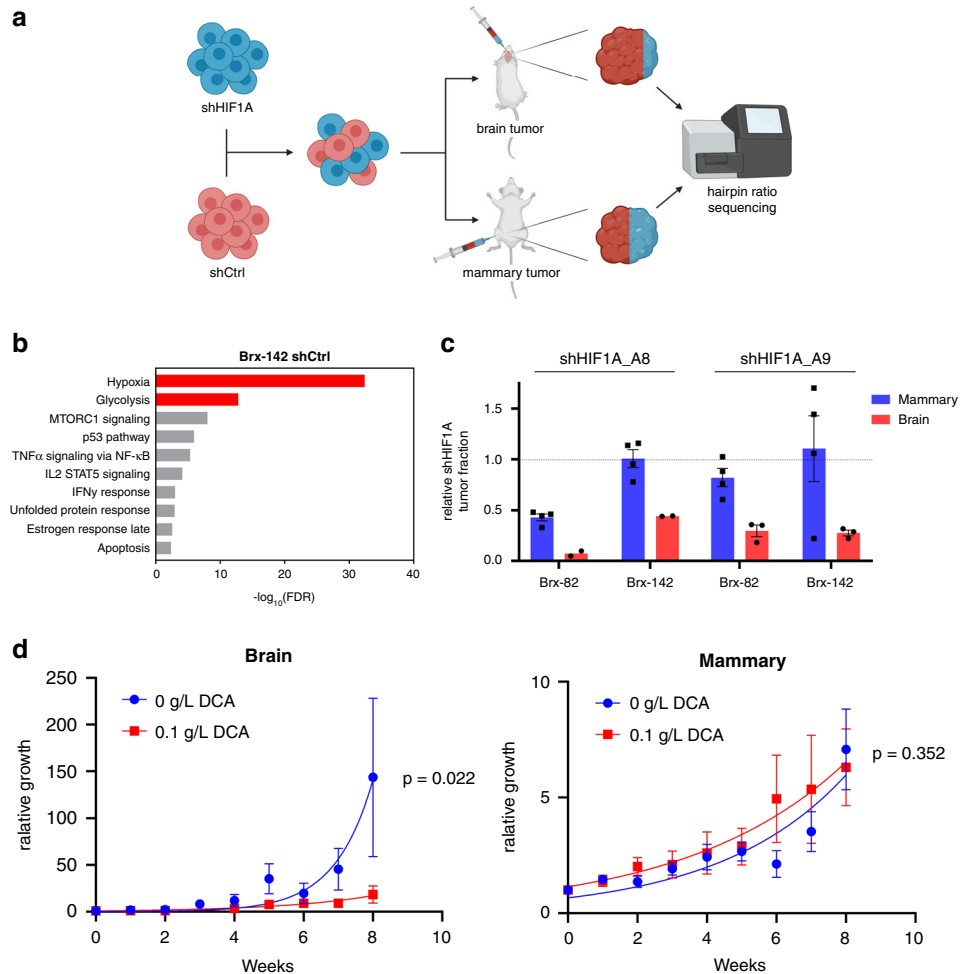

**Fig. 5 Hypoxic signaling is preferentially required for brain versus breast tumor growth. a** Schema illustrating mixing experiments using shCtrl and shHIF1A CTCs to establish brain and mammary tumors. **b** GSEA of transcripts differentially expressed in Brx-142 shCtrl versus shHIF1A_A8 cells (fold change >2; FDR < 0.25). The most enriched Hallmarks of Cancer gene sets from the Broad Molecular Signatures Database and associated FDR values are shown. Hypoxia and glycolysis genesets are highlighted. **c** shHIF1A tumor fraction as compared to input fraction in brain or breast tumors from Brx-82 or Brx-142 mixing experiments. shHIF1A tumor fraction was determined via next-generation sequencing for the hairpin sequences corresponding to shCtrl, shHIF1A_A8, and shHIF1A_A9 (shHIF1A_A8 brain: $n = 2$; shHIF1A_A9 brain: $n = 3$; all mammary: $n = 4$). Dotted line indicates unchanged shHIF1A and shCtrl fractions from input. **d** Brain or mammary tumor luminescence monitored following stereotactic injection of Brx-82 cells (brain 0 g/L DCA: $n = 8$; brain 0.1 g/L DCA: $n = 4$; all mammary: $n = 4$). Mice were treated for 8 weeks with 0.1 g/L DCA or vehicle delivered in drinking water. *P* values calculated by the extra sum-of-squares F test. Data for in vitro experiments represent mean ± SD and for in vivo experiments represent mean ± SEM. Source data are provided as a Source Data file.

glycolytic signaling, consistent with the role of HIF1A in promoting anaerobic metabolism (Fig. 5b and Supplementary Figs. 12 and 13). Following tumor engraftment of the mixed shHIF1A and shCtrl cell populations, the fraction of shHIF1A cells was dramatically reduced in brain compared with mammary tumors (mean 0.31; 95% CI: 0.18-0.43). The decreased fraction of shHIF1A cells in brain tumors was observed for both Brx-82 and Brx-142 CTC lines (Fig. 5c). In contrast, in the majority of mammary tumor samples, the fraction of shHIF1A cells was unchanged from input, suggesting that HIF1A has minimal effect on the growth of orthotopic mammary tumors in this model. Thus, HIF1A is preferentially required for the growth of patient-derived breast CTCs in the brain versus mammary gland.

The increased HIF1A expression in brain-proficient F1 CTC lines is associated with transcriptional and metabolic changes indicating a shift towards increased anaerobic metabolic activity. We therefore tested whether pharmacological inhibition of anaerobic metabolism exerts a differential effect on the growth of CTC lines in the brain versus mammary gland. Cultured CTCs

were treated with dichloroacetic acid (DCA), a pyruvate dehydrogenase kinase inhibitor known to enhance pyruvate transport to the mitochondria and promote a metabolic shift toward increased oxidative phosphorylation[31]. Consistent with this effect, DCA-treatment of CTCs in vitro leads to decreased anaerobic metabolism and increased oxidative phosphorylation (Supplementary Fig. 14). To test the effects of DCA treatment in vivo, we injected Brx-82 CTCs into either the brain or mammary gland of recipient mice and treated the mice with either oral DCA (0.1 g/L) or vehicle for 8 weeks. Brain tumor growth was significantly reduced following treatment with DCA ($P = 0.022$) (Fig. 5d). In contrast, mammary tumor growth was unchanged following treatment with DCA ($P = 0.352$). Taken together, these data again suggest that HIF1A is preferentially required for the growth of breast cancer-derived CTCs in the brain.

**CTC hypoxic signaling predicts poor clinical outcome in brain metastasis patients.** Finally, in order to assess the role of hypoxic

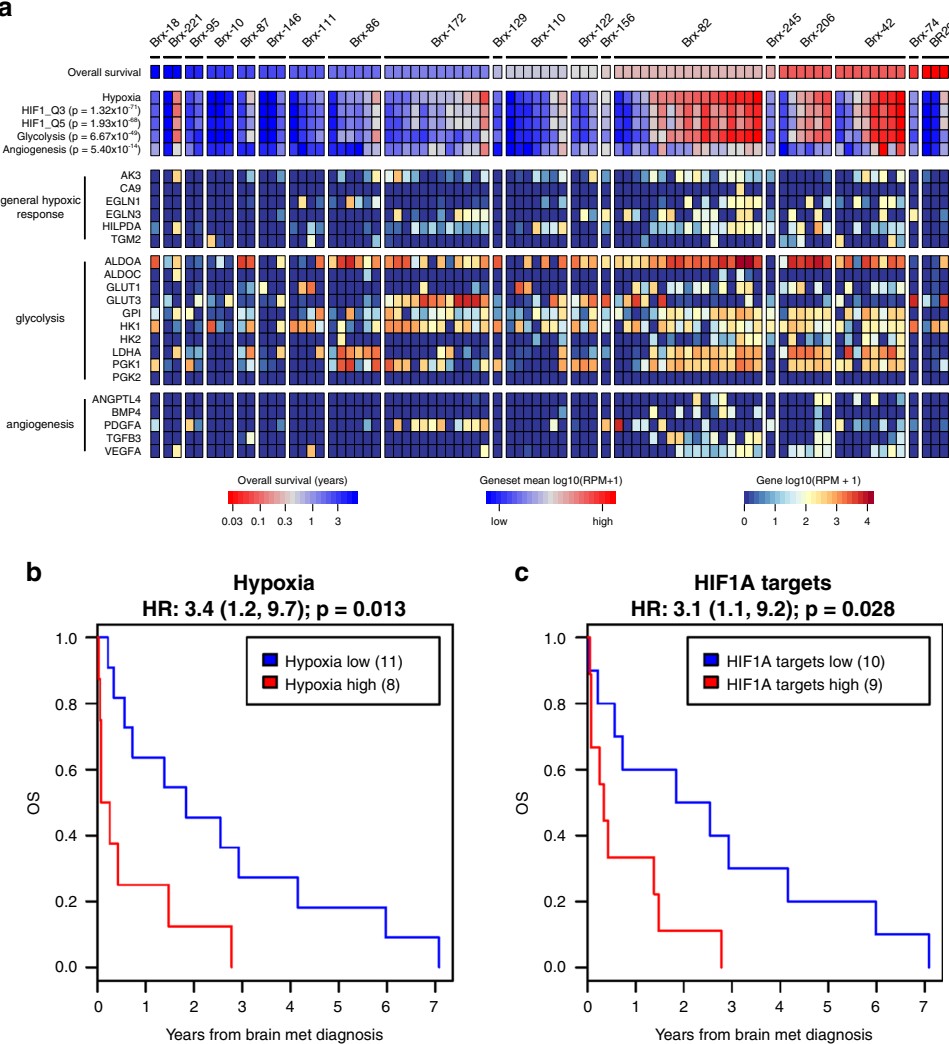

**Fig. 6 Hypoxic signaling in CTCs predicts poor outcome in brain metastasis patients. a** RNA-seq from CTCs enriched from whole blood of breast cancer patients with brain metastases using the iChip microfluidic device[24,25,32]. Heat map of the expression level of canonical HIF1A target genes. The color bars illustrate metagene analysis of hypoxia, HIF1A targets, glycolysis and angiogenesis signatures. Patients ordered by decreasing overall survival (OS) following brain metastasis diagnosis. Per-patient CTCs ordered by increasing hypoxia expression. **b** Kaplan–Meier analysis of the OS following brain metastasis diagnosis for patients with high average hypoxia gene expression in CTCs versus those with low average hypoxia gene expression (Hallmark Hypoxia geneset). The hypoxia-high and hypoxia-low subgroups were determined on the basis of average hypoxia gene expression across all CTCs isolated for each patient. *P* value calculated by log rank test. **c** Kaplan–Meier analysis of the OS following brain metastasis diagnosis for patients with high average HIF1A target gene expression in CTCs versus those with low average HIF1A target gene expression (Transcription Factor Targets HIF1_Q5 geneset). The HIF1A targets-high and HIF1A targets-low subgroups were determined on the basis of average HIF1A target gene expression across all CTCs isolated for each patient. *P* value calculated by log rank test.

and HIF1A signaling in primary CTCs, we interrogated RNA-seq profiles of 83 freshly isolated single cell CTCs or CTC clusters from 19 breast cancer patients with known brain metastases[24,25,32] (Supplementary Data 6). This cohort includes patients with HR+, HER2-amplified, and TNBC breast cancer subtypes. Transcriptomic analysis demonstrates a range of hypoxia-related signaling in individual patient CTCs, with a >10-fold range in mean expression of hypoxic genes and HIF1A target genes (Fig. 6a). As in our in vitro and in vivo models, increased hypoxic signaling in CTCs significantly correlates with increased glycolytic and angiogenic signaling (glycolysis: $P = 6.67 \times 10^{-49}$; angiogenesis: $P = 5.4 \times 10^{-14}$). We collected clinical outcome data for the patients, allowing for correlation of CTC RNA-seq data with patient overall survival (OS) following diagnosis of brain metastasis. To account for variations in number of CTCs isolated per patient, we assessed mean expression of hypoxia genes across all CTCs collected from

each patient. This per-patient analysis reveals that increased expression of hypoxia genes in CTCs is associated with significantly reduced OS (HR: 3.4, $P = 0.013$) (Fig. 6b), as is increased expression of HIF1A targets (HIF1_Q3: HR: 2.6, $P = 0.056$; HIF1_Q5: HR: 3.1, $P = 0.028$) (Fig. 6c and Supplementary Fig. 15). Importantly, these correlations of CTC hypoxic signaling to OS persist after controlling for ER, PR, and HER2 status and thus are independent of breast cancer subtype (Supplementary Fig. 16). That increased hypoxic and HIF1A signaling in CTCs predicts poor clinical outcome in breast cancer patients with brain metastases further highlights the importance of these pathways in brain metastasis progression and confirms the clinical relevance of our findings.

## Discussion

Using patient-derived breast CTC cultures in mouse models, we have uncovered a differential role for HIF1A-dependent signaling

in brain metastasis. CTC lines serially enriched for their ability to proliferate in the brain demonstrate increased expression of hypoxia-related pathways. In mouse and human, nuclear expression of HIF1A itself is increased within breast cancer brain metastases, and in cell mixing experiments, suppression of HIF1A preferentially suppresses proliferation in the brain. Finally, HIF1A signaling within CTCs of patients with brain metastases predicts poor clinical outcome. Taken together, these results suggest that HIF1A signaling may play an important role in the proliferation of breast cancer cells in the brain.

Cancer metastasis has long been thought to encompass multiple distinct cellular properties, both at the level of the cancer cell (seed) and the microenvironment (soil). Tissue-specific homing of metastatic cancer cells may be evident in some cancers[33], and most experiments on brain tropism have focused on genes that promote the ability of tumor cells to cross the BBB[4–6,34,35]. However, in models involving direct intracardiac injection, metastatic cancer cells appear to be initially widely distributed throughout the animal, but selectively proliferate in only a subset of tissues, pointing to the importance of tissue-specific proliferation signals[4,6]. Our use of direct brain injections was therefore designed to identify specific requirements for growth within the brain environment, bypassing survival of cancer cells in the circulation and migration through the BBB. Interestingly, several genes previously shown to enhance brain metastatic capacity in models based on intravascular injection include immediate downstream effectors of HIF1A signaling, such as VEGF and L1CAM[19,34].

HIF1A signaling mediates the cellular response to hypoxia, enhancing glycolysis, angiogenesis, and EMT, among other tumor-associated phenotypes[13]. Given these effects, HIF1A has been implicated in multiple cancer types, with recent studies uncovering a role of primary tumor hypoxia as an initiating factor in the metastatic cascade, promoting extracellular matrix remodeling[36], microvesicle release[37], and immune suppression[38], among others. Complementing these insights into early invasion, we now highlight the role of HIF1A in brain-specific proliferation. Activation of HIF1A signaling may occur within some cells in the primary tumor or in other metastatic sites, generating CTCs with increased ability to proliferate in the brain. Such a model is supported by the increased HIF1A signaling in patient blood-derived CTC lines with high growth potential in the brain, as well by the heterogeneity of freshly isolated single CTCs from women with metastatic breast cancer. In this case, the serially enriched brain-competent isogenic CTCs may reflect selection pressure for pre-existing cells with elevated hypoxic signaling, although we cannot exclude additional induction of HIF1A activity during the process of brain tumor initiation. Recent studies have suggested that hypoxic signaling initiated in primary tumor cells may be maintained through epigenetic mechanisms in the circulation and following dissemination[39,40], a concept that is supported by our observation of preserved hypoxic activity in the different brain-competent CTC lines cultured in vitro.

Despite the brain consuming 20% of whole body oxygen, brain metastases may demonstrate reduced partial pressures of oxygen (pO$_2$) versus primary tumors[11], with a median pO$_2$ of 10 mmHg in a meta-analysis of primary breast tumors, but a median pO$_2$ of 4.4 mmHg in a cohort of breast cancer brain metastases; brain metastases also demonstrate increased proportions of severely hypoxic regions versus primary breast tumors[41,42]. Normal neurons express the oxygen carrier neuroglobin, which binds oxygen with an affinity higher than that of hemoglobin[43], and may thus further deplete the oxygen available to cancer cells lacking this oxygen sequestration mechanism. Combined, these factors suggest that breast cancer brain metastases face both more profound and more widespread hypoxia versus primary tumors, potentially contributing to their increased dependence on HIF1A signaling for proliferation.

Cell mixing experiments, involving co-injection of HIF1A knockdown and control cells, demonstrated a striking reduction in brain proliferation by shHIF1A cells, with minimal effect on mammary tumor growth. The lack of effect by HIF1A knockdown on mammary tumors contrasts with previous studies describing HIF1A promotion of growth in primary breast tumors[44,45]. This difference may reflect our use of CTCs, which are metastatic precursor cells, compared with ATCC cell lines derived from primary breast cancers. Alternatively, it is also possible that, in our mammary cell mixing experiments, HIF1A-mediated release of growth factors and cytokines from control cells supports mammary tumor growth of cocultured cells with HIF1A knockdown through paracrine mechanisms[46,47]. In the brain cell mixing experiments, however, HIF1A knockdown clearly contributes to cell autonomous, intrinsic cell viability, and proliferation, as evidenced by both our cell mixing and DCA drug studies. Importantly, while our experiments were carried out using HR+ CTC lines, analysis of a brain-tropic TNBC cell line demonstrates enrichment of hypoxic signaling, and CTC hypoxia predicts metastatic disease progression independent of subtype in a cohort comprising patients with HR+, HER2-amplified, and TNBC tumors; as such, our findings may be applicable across diverse breast cancer subtypes.

Finally, our observations have several implications for the treatment of breast cancer brain metastases. First, elevated hypoxic signaling suggests a possible explanation for the common failure of systemic cancer therapies in brain metastases[48]. While therapeutic resistance is generally thought to result from poor penetration through the BBB, brain metastases often compromise this barrier with evident vascular leakage[49], and even small molecules with good brain penetrance often demonstrate lower efficacy against brain metastases compared with other sites of disease[50]. Given the known effect of hypoxia and HIF1A signaling pathways in mediating resistance to various targeted therapies[51], our observations raise the possibility that increased hypoxic signaling may also contribute to the common failure of systemic therapies on intracranial metastases. Second, the correlation of increased hypoxic signaling with poor outcome in brain metastasis resection specimens and in CTCs collected from patients with brain metastases raises the possibility of hypoxia-targeted therapies in these patients. Recent clinical studies have explored the use of anti-angiogenic drugs in the treatment of breast cancer brain metastases[20,52]; our data suggest that inhibition of additional downstream pathways of hypoxic signaling, including glycolysis, or even inhibition of hypoxic signaling itself may slow progression of brain metastatic disease. HIF-targeting therapies that are currently under development may be considered to overcome treatment resistance or to slow metastatic proliferation in the brain.

## Methods
**CTC culture.** CTCs were cultured in 4% O$_2$ in suspension conditions using ultra-low attachment plates (Corning) in media consisting of RPMI-1640 with Gluta-MAX supplemented with EGF (20 ng/mL), FGF (20 ng/mL), 1X B27, and 1X antibiotic/antimycotic (Life Technologies)[23,24,53]. CTCs were checked for mycoplasma (MycoAlert, Lonza), and were authenticated by RNA-seq and DNA-seq.

**Stereotactic brain and mammary fat pad injections.** Mice were housed in a specific pathogen-free environment in the animal facility at the Massachusetts General Hospital Cancer Center, and all experiments conformed to ethical principles and guidelines approved by the Institutional Animal Care and Use Committee of the Massachusetts General Hospital (Protocol 2010N000006). After receiving isoflurane anesthesia and buprenorphine analgesia, 6-week-old female NSG mice (NOD. Cg-Prkscsdid Il2rgtm1Wjl/SzJ) from Jackson Laboratories were stereotactically injected with $5 \times 10^5$ cells per mouse in the right frontal lobe of the brain or orthotopically injected with $2.5 \times 10^5$ cells in the right or left fourth mammary fat pad. For brain injections, the cranial burrhole was created 2.5 mm to the right of the bregma on the coronal suture, and a Hamilton syringe (Hamilton)

was inserted to a depth of 2.5 mm below the outer table of the calvarium using a stereotactic injection system[54]. A 90-day release 0.72 mg estrogen pellet (Innovative Research of America) was implanted subcutaneously behind the neck of each mouse. Postoperatively, mice received buprenorphine analgesia twice daily for three days. Thereafter, mice were monitored for signs of pain or neurological dysfunction at least daily and were treated with buprenorphine for pain or sacrificed in the setting of neurologic dysfunction. Tumor growth was monitored weekly via in vivo imaging using the IVIS Lumina II (PerkinElmer) following intraperitoneal injection of D-luciferin (Sigma). To generate F1 and F2 cultures, brain tumors were digested with collagenase/hyaluronidase at 37 °C, washed, and re-cultured in vitro with growth conditions the same as those for parental cells, as described above[23]. After expansion, cells were live-sorted for GFP using a Laser BD FACS Aria Fusion Cell Sorter, BSL2+. F1 and F2 cultures were grown in vitro for a maximum of 2 months.

**Histology and immunohistochemistry**. Tumors were fixed in 10% formalin for 24 h, followed by preservation in 70% ethanol. Tissue was paraffin embedded and cut into 5 μm sections. Sections were stained with hematoxylin and eosin, or immunohistochemical staining was performed[53]. Following permeabilization, antigen retrieval, and blocking, sections were incubated with primary antibodies against GFP (1:250; Abcam ab183734), Ki-67 (1:50; Life Technologies 180192Z), Cleaved caspase-3 (1:1000; Cell Signaling Technology 9664S), or HIF1A (1:1000; Novus NB100-131) for 1 h at room temperature. Sections were incubated with HRP anti-rabbit antibody (DAKO) for 30 min, then incubated with 3,3′-diamino-benzidine (Vector Laboratories) for 10 min. Sections were counter-stained with Gill #2 hematoxylin for 10–15 s. Stained tissue sections were digitized using the Aperio CSO (Leica Biosystems). Quantification of cells positive for Ki-67 or cleaved caspase 3 was performed manually using Aperio ImageScope software. Tumor area was defined by GFP staining of serial sections.

**In vitro growth**. About 2000–5000 CTCs were seeded in tumor sphere media in 96-well ultra-low attachment plates (Corning) in quadruplicate. Cell viability was assayed with CellTiter-Glo (Promega) and was normalized to day 0 signal.

**Quantitative real time PCR**. RNA was isolated using RNeasy Mini Kits (Qiagen). RNA was reverse transcribed using Superscript III First Strand Synthesis Supermix (Invitrogen), and qRT-PCR was performed using TaqMan probe and primer sets for HIF1A and ACTB (ThermoFisher Biosciences) (Supplementary Data 7). Values represent the ratio of the relative quantity of HIF1A transcript to the relative quantity of ACTB transcript.

**Oncogene and tumor suppressor DNA sequencing**. Genomic DNA was isolated from Brx-50 and Brx-142 parental and F1 cells, and from Brx-82 parental, F1 and F2 cells. The genomic DNA was enzymatically sheared, end-repaired, adenylated, and ligated with a half-functional adapter. A sequencing library targeting hotspots and full exons was generated using two hemi-nested PCR reactions[27]. Illumina NextSeq paired-end sequencing results were aligned to the hg19 human genome reference using Novoalign. An ensemble variant calling approach was applied for SNV and indel variant detection. A copy number caller utilizing a coverage distribution from a panel of normals was applied for copy gain and loss detection.

**RNA-seq library generation and sequencing**. Amplified cDNA was generated using 10 ng RNA from each sample using the SMART-Seq HT Kit (Takara Bio) according to manufacturer protocol. Briefly, first-strand synthesis was performed using oligo-dT primers followed by template switching by the reverse transcriptase, second strand synthesis, and 18 cycles of amplification. Amplified cDNA was purified with 1× bead cleanup with Agencourt AMPure XP beads (Beckman Coulter). The Nextera XT DNA Library Preparation kit (Illumina) was used for library generation with 1 ng cDNA input for enzymatic tagmentation, followed by 12 cycles of amplification and addition of unique dual-index barcodes. PCR product was purified with 1.8X bead cleanup. After qPCR-based quantification using the KAPA Library Quantification kit (Roche), individual libraries were pooled and subsequently sequenced on a NextSeq 500 system (Illumina) using a 150 cycle V2.5 high output kit with paired end-read mode.

**Determination of RNA-seq reads-per-million (RPM)**. Trimmomatic was used to crop read lengths to 50 nucleotides and to remove the TruSeq3-PE-2 Illumina adapters. The reads were then aligned using tophat2 and bowtie1 with the no-novel-juncs argument set with human genome version hg19 and transcriptome defined by the hg19 genes.gtf table from http://genome.ucsc.edu. Reads that did not align or aligned to multiple locations were discarded. For specimens that were a tumor from a xenograft, the reads were also aligned in the same way to the mouse genome and transcriptome version mm10. Reads that aligned to the mouse transcriptome were removed from the collection of reads that aligned to the human transcriptome. The number of remaining reads aligning to each gene in the human transcriptome was then determined using htseq-count. The read count for each gene was divided by the total counts assigned to all genes and multiplied by one million to form the reads per million (RPM).

**Gene set enrichment analysis of RNA-seq**. Differential gene expression was determined as follows. First, genes with 90th quantile of RPM values less than 10 were discarded. Then we used the classic mode of the Bioconductor edgeR package with common dispersion set to 0.01, the recommended setting for genetically identical model organisms. Genes for which the fold-change in either direction as determined by edgeR was ≥2 and for which the FDR estimate determined by edgeR was ≤0.25 were considered differentially expressed. To identify gene set enrichment, a hypergeometric test was then performed looking for enrichment of differentially expressed genes in the HALLMARK gene-set collection of version 6.0 of the Broad Institute's MSigDB. We also looked for enrichment in this gene-set collections using the Broad Institute's GSEA software in pre-ranked mode, giving as input for each gene -$\log_{10}(P$ value)*[−1 if higher in control or parental; 1 if higher in treated or F1].

**Seahorse metabolite assessment**. Extracellular acidification rates and oxygen consumption rates were determined using a Seahorse XFE96 Analyzer (Agilent). Samples were prepared and run on the XFE96 Analyzer per manufacturer's instructions; 50,000 cells were used per sample, and 10 replicates were tested per condition.

**Metabolomics**. Cells were washed once with ice cold 0.9% NaCl and extracted on dry-ice in 1 mL 80% methanol containing 500 nM internal standards (Metabolomics Amino Acid Mix Standard: Cambridge Isotope Laboratories, Inc.). Cell extract was collected using a cell scraper and transferred to a microcentrifuge tube. Samples were vortexed for 10 min at 4 °C and centrifuged at 17,000 × g for 10 min at 4 °C. The supernatant was transferred to a new tube and evaporated to dryness under nitrogen. Dried polar extracts were stored at −80 °C until analysis.

Metabolite profiling was conducted on a QExactive bench top orbitrap mass spectrometer equipped with an Ion Max source and a HESI II probe, which was coupled to a Dionex UltiMate 3000 HPLC system (Thermo Fisher Scientific, San Jose, CA). External mass calibration was performed using the standard calibration mixture every 7 days. Typically, dried polar fractions were reconstituted in 100 μL water and 2 μL were injected onto a SeQuant ZIC-pHILIC 5 μm 150 × 2.1 mm analytical column equipped with a 2.1 × 20 mm guard column (MilliporeSigma). Buffer A was 20 mM ammonium carbonate, 0.1% ammonium hydroxide; Buffer B was acetonitrile. The column oven and autosampler tray were held at 25 and 4 °C, respectively. The chromatographic gradient was run at a flow rate of 0.150 mL/min as follows: 0–20 min: linear gradient from 80% to 20% B; 20–20.5 min: linear gradient form 20% to 80% B; 20.5–28 min: hold at 80% B. The mass spectrometer was operated in full-scan, polarity-switching mode, with the spray voltage set to 3.0 kV, the heated capillary held at 275 °C, and the HESI probe held at 350 °C. The sheath gas flow was set to 40 units, the auxiliary gas flow was set to 15 units, and the sweep gas flow was set to 1 unit. MS data acquisition was performed in a range of $m/z = 70$–1000, with the resolution set to 70,000, the AGC target at $1 \times 10^6$, and the maximum injection time at 20 ms. An additional scan ($m/z = 220$–700) was included in negative mode only to enhance detection of nucleotides. Relative quantitation of polar metabolites was performed with XCalibur QuanBrowser 2.2 (Thermo Fisher Scientific) using a 5 ppm mass tolerance and referencing an in-house library of chemical standards.

**Nuclear HIF1A quantitation**. Image quantification was performed using Halo software (Indica Lab) Multiplex IHC module. Tumor regions were hand annotated and validated by a trained pathologist. For each tissue section, the annotated tumor region(s) contained more than 1000 cells, with the exception of one breast sample, which had 262 cells. Color deconvolution was used to separate hematoxylin and HIF1A stain color vectors so that each individual cell nucleus could be segmented and HIF1A staining intensity could be quantified. Nuclear segmentation was based on hematoxylin stain color vector. For each cell object, HIF1A nuclear staining intensity was recorded in optical density units (OD). To ease comparison between breast and brain samples, all HIF1A intensity measurements were normalized by the mean breast nuclear intensity per sample.

Based on HIF1A nuclear intensity values across all patient samples analyzed, a global threshold was chosen for HIF1A positivity. The percentage of HIF1A positive cells relative to the total number of cells analyzed per sample was determined. The same threshold and analysis algorithm was applied to mouse tissue sections.

**Lentivirus production and transduction**. HEK293T cells were grown in high-glucose DMEM with 10% fetal bovine serum and 1% penicillin/streptomycin. HEK293T cells were transfected using pMD2.G and psPAX2 packaging plasmids, and viral supernatant was harvested 48 h post-transfection and filtered through a 0.45 μm PVDF filter. CTCs were transduced with lentivirus with 6 μg/mL Polybrene for 24 h[53]. After 72 h of infection, CTCs were selected using puromycin (3 μg/mL) for 7 days.

**Mixing experiments**. Brx-82 and Brx-142 CTCs expressing shCtrl and CTCs expressing either shHIF1A_A8 or shHIF1A_A9 (Supplementary Data 7) were mixed at a 1:1 ratio and injected into mouse right frontal lobe or mammary fat pad, as described above. Tumor growth was monitored weekly via in vivo imaging as

above, and tumors were harvested when they had grown to 100 times the original injection bioluminescent signal. Tumors were divided into 25 µg chunks and homogenized using a TissueLyser II (Qiagen), and DNA was extracted using NucleoSpin Tissue DNA extraction columns (Macherey-Nagel). PCR of the guides was performed using NEBNext High Fidelity 2X Master Mix (New England Bio-labs) in parallel reactions in a single-step reaction of 36 cycles, using primers designed to amplify the small hairpin sequence (Supplementary Data 7). PCR products were purified via SPRI bead cleanup, pooled, and sequenced on the Illumina MiSeq platform.

Processing of sequencing data to enumerate the fraction of each shRNA was performed using R-Studio. Fastq files obtained after sample deconvolution using Illumina BaseSpace software were processed using the processAmplicons function in edgeR within Bioconductor[55]. Parameters included allowing up to four mismatches within the shRNA. The ratio of reads for each HIF1A shRNA to reads for the control shRNA was calculated.

**DCA treatment.** Brx-82 CTCs were injected into mouse right frontal lobe or mammary fat pad, as described above. Following injection, half of each cohort received water with 0.1 g/L dichloroacetic acid (DCA) added, while the other half of each cohort received water with vehicle. Water with DCA or vehicle was replen-ished weekly. Brain or mammary tumor growth was measured weekly via in vivo imaging using the IVIS Lumina II (PerkinElmer) following intraperitoneal injec-tion of D-luciferin (Sigma).

**CTC isolation from patients with breast cancer brain metastases.** Patients with a diagnosis of metastatic breast cancer with brain metastases provided informed consent for de-identified blood and clinical data collection, as per institutional review board approved protocol (DF/HCC 05-300) at Massachusetts General Hospital. Samples of ~6–12 mL of fresh whole blood were processed through the microfluidic CTC-iChip[24,25,32,56]. Briefly, to magnetically label white blood cells, whole blood was incubated with biotinylated antibodies against CD45 (R&D Systems, clone 2D1) and CD66b (AbD Serotec, clone 80H3), followed by incu-bation with Dynabeads MyOne Streptavidin T1 (Invitrogen). Samples were then passed through the CTC-iChip. CTCs in CTC-iChip product were identified via staining with Alexa Fluor 488-conjugated antibodies against EpCAM (Cell Sig-naling Technology, #5198), Cadherin 11 (R&D Systems, FAB17901G), and HER2 (BioLegend, #324410). Contaminating white blood cells in CTC-iChip product were identified via staining with TexasRed-conjugated antibodies against CD45 (BD Biosciences, BDB562279), CD14 (BD Biosciences, BDB562334), and CD16 (BD Biosciences, BDB562320). Single CTCs were identified based on intact cellular morphology, Alexa Fluor 488-positive staining, and lack of TexasRed staining.

**Statistics and reproducibility.** All statistical tests used are noted in figure legends. All *n* indicated in figures represent independent experimental samples and not technical replicates. All data generated for each experimental condition were included, with the exception of inaccurate in vivo growth values resulting from failed intraperitoneal D-luciferin injections, which were censored. All statistical tests were two-tailed, and P values ≤0.05 were consider statistically significant. For all growth analyses, in vivo and in vitro, statistical differences in growth rate models were calculated using extra sum-of-squares F tests. For Kaplan–Meier analysis of mice inoculated with parental, F1, or F2 cells, statistical differences in survival were calculated by log rank tests. Kaplan–Meier analysis of patients with high hypoxia signature CTCs versus patients with low hypoxia signature CTCs was conducted as follows. For each of the patients in the two datasets (GEO GSE144494 and GEO GSE144495) we computed the mean log10(RPM + 1) value of the genes in the hypoxia signature and averaged those means across all the CTCs from that patient. For each dataset, we classified those averages as high or low using Otsu's method[57]. We then made a Kaplan–Meier plot and performed the log rank and Cox proportional hazards tests using the patients from both datasets who had brain metastases and for whom we could obtain overall survival data (Supplementary Data 6). The same procedure was followed for the HIF1A targets signatures. For comparison of HIF1A+ cells in mouse or human brain or mammary samples, statistical differences in percentages of HIF1A+ cells were calculated using two-tailed two population proportion z-tests. For all other comparisons—including comparisons of glycolytic activity, oxidative phosphorylation, HIF1A nuclear staining quantity in mouse or human brain or mammary samples, and relative shHIF1A tumor fractions in mixing experiments—statistical differences were cal-culated by two-tailed unpaired t-tests.

Statistical analysis of enrichment of genesets was conducted using GSEA, as described in the "*Gene set enrichment analysis of RNA-seq*" methods section. For correlation of geneset enrichment in patient CTC data, Pearson correlation between Hallmark Hypoxia and TFT HIF1_Q3, TFT HIF1_Q4, Hallmark Glycolysis, and Hallmark Angiogenesis was computed separately for each of the two datasets (GEO GSE144494 and GSE144495) and then combined using the metacor function of the R package meta. In so doing, when the heterogeneity P value was <0.05, we used the random effects model; otherwise, we used the fixed effect model.

**Reporting summary.** Further information on research design is available in the Nature Research Reporting Summary linked to this article.

## Data availability
Raw data from RNA-seq of CTC cultures and CTC-derived mouse xenograft mammary and brain tumors have been deposited in the Gene Expression Omnibus (GEO) database under accession number GSE156944. Source data are provided with this paper (Figs. 1–5 and Supplementary Figs. 1, 2, 3, 6, 8, 11, and 14), and all data and materials are available from the corresponding authors upon request. A reporting summary is available as a Supplementary Information file.

## Code availability
Project-specific computer code is available at GitHub [https://github.com/richardebright/HIF1A]. All other computer code is available upon request.

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

## Acknowledgements
We are grateful to all the patients who participated in this study. We thank L. Libby for technical support and C. Lewis for expert metabolite profiling. Figures created with BioRender.com. This work was supported by NIH Grant 2R01CA129933, the Breast Cancer Research Foundation, the Howard Hughes Medical Institute, and the National Foundation for Cancer Research (D.A.H.), NIH Quantum Grant 2U01EB012493 (M.T., D.A.H.), NIH Grant U01CA214297 (M.T., D.A.H., S.M.), ESSCO Breast Cancer Research (S.M.), Breast Cancer Research Foundation Grant ELFF-18-003, Damon Runyon Cancer Research Foundation Grant 81C-15, NIH Grants 1R01CA244975-01, 5R01CA227156-02, 5R21CA220253-02 (P.K.B.), T32GM007753, 1F30CA232407-01 (R.Y.E.), American Cancer Society 132140-PF-18-127-01-CSM, ASCO Young Investigator Award (D.S.M.), Neurosurgery Research and Education Foundation Fellowship, AMA Seed Grant (M.A.Z.).

## Author contributions
R.Y.E., M.A.Z., D.A.H., and S.M. conceived the project and provided project leadership. B.S.W., L.T.N., and B.C. performed bioinformatics analyses. D.S.M., K.L.N., D.F.W., and B.W. assisted with molecular biology and animal experiments. B.S., E.N.B., L.A., and P.K.B. collected, annotated and processed clinical samples. A.S. and D.T.T. provided RNA-seq support. A.J.I. provided DNA sequencing support. M.T. collaboratively developed the CTC-iChip isolation of viable CTCs.

## Competing interests
D.T.T., M.T., D.A.H., and S.M. are founders of and own equity in TellBio, Inc., which is involved with CTC therapeutics and diagnostic. At this time, there has been no funding received or license that has been given to TellBio, Inc. for this work. D.T.T. is also a founder and owns equity in ROME Therapeutics and PanTher Therapeutics, which is not related to this work. D.T.T. has received consulting fees from ROME Therapeutics, Foundation Medicine, Inc., NanoString Technologies, EMD Millipore Sigma, and Pfizer that are not related to this work. D.T.T. receives research support from ACD-Biotechne, PureTech Health LLC, Ribon Therapeutics, which was not used in this work. D.T.T.'s interests were reviewed and are managed by Massachusetts General Hospital and Mass General Brigham in accordance with their conflict of interest policies. P.K.B. has received research support (to Massachusetts General Hospital) from Merck, Pfizer, Lilly, and BMS. P.K.B. has consulted for Genentech-Roche, ElevateBio, Lilly, and AngioChem. P.K.B. has received Speaker's Honoraria from Genentech-Roche and Merck. R.Y.E. has received consulting fees from Nextech Invest and nference Inc., which are not related to this work. The other authors declare no competing interests.
