## [Peer Review File · Nature Communications]

REVIEWER COMMENTS

Reviewer #1 (Remarks to the Author):

The manuscript “Hypoxia signaling selectively promotes proliferation of breast cancer metastases in the brain” describes an innovative approach where CTCs from breast cancer patients are harvested from blood samples and maintained in suspension culture in vitro. Cells from these CTCs’ cultures are then implanted in the brain of NSG mice to evaluate their propensity to grow in the brain microenvironment. The authors select 3 of the CTC cell lines with moderate to slow growth potential and enrich these cells by serial intracranial injections. By performing RNA sequencing, the authors conclude that hypoxia signaling is enriched in the derivative cell lines. By performing IHC on matched primary tumor and brain metastases, the authors show a significant increase of HIF1A staining in the brain compared to the lung. Finally, the authors silence HIF1A expression (shRNA), mixed these cells with a scramble control, and demonstrated reduced fraction of shHIF1A cells in brain tumors versus unaffected growth in mammary tumors. There are many publications which the authors should reflect on (by citing) that speak to the role of hypoxia in breast cancer metastasis and the role of HIF1. The novelty of the current study is that serial passage of CTCs in the brain by intracranial injection promotes enhances the ability to proliferate and enriches for a hypoxia signaling program.

Major concerns:

1. “Hypoxia signaling selectively promotes proliferation of breast cancer metastases in the brain” – title Data does not support title. The study does not examine the role of hypoxia in metastasis. Instead, the experiments show that HIF1A is required for proliferation of breast cancer cells when injected directly into the brain microenvironment. A spontaneous model in which cells exposed to hypoxia in the primary tumor could be “followed” and tested for their ability to spontaneously metastasize would be required to support the title.
2. The first experiment is well described and compares the proliferative capacity of 7 distinct patient-derived CTCs in the brain of NSG mice post intracranial injection. The authors highlighted 2 rapid-, 2-moderate and 3-slow proliferating cell lines. They then show images of Ki67 and cleaved Casp3 staining in BRx-82 (moderate-growth cell line and mistakenly indicated as rapid in the text (typo)) versus BRx-142. I am not sure what the goal of this experiment is or how it fits into the context of the rest of the paper which aims to study hypoxic signaling as a mechanism for proficeincy to grow in the brain. Do some cell lines have a greater enrichment for the hypoxia signaling that others? Why did the authors choose to continue with 50, 82, and 142 and not the other CTC lines?

3. Please clarify the methods used for the the handling conditions of the cells prior to RNA sequencing. If the cells were cultured ex vivo, the experiment suggests that gene expression in the F1 versus parental cell lines that are driven by exposure to the brain niche can be maintained ex vivo? How does this occur? This is an interesting finding and would be supported by the findings of Godet, et al. Nat Commun 10, 4862 (2019). Godet et al shows that non-genomic changes that occur under hypoxia in vivo can be maintained when the cells are returned to tissue culture implying a “hypoxic memory”.

4. It is unclear whether the CTCs have an “preprogrammed” propensity for increased proliferation in the brain or whether the brain microenvironment reprograms the CTCs. Given that hypoxic signaling is preferentially required for growth in the brain versus the primary site, one could argue that the hypoxia-program only has a benefit in the brain niche. On the other hand, the difference in HIF-1 staining in the brain versus the primary tumor suggests that the hypoxic program is only ‘turned on’ when the cells are localized in the brain. In order to reconcile this question, the authors need to perform sequencing on freshly resected cells sorted from the brain versus the primary tumor. This is critical to determine whether or not hypoxic cells originating in primary tumor have a pre-programmed propensity to survive/proliferate in the brain microenvironment. This would really enhance the novelty of the study.

5. To confirm the role of hypoxia signaling on survival and proliferation in the brain, the authors injected a mixture of shHIF1A and shCtrl CTC cells into the brain or into the mammary fat pad. They then show that the fraction of shHIF1a cells was reduced in the brain at the endpoint of the experiment. This was not observed in the mammary tumors. There have been many studies linking primary tumor growth and HIF1A. Please see: Schwab, L.P., Peacock, D.L., Majumdar, D. et al. Hypoxia-inducible factor 1 α promotes primary tumor growth and tumor-initiating cell activity in breast cancer. Breast Cancer Res 14, R6 (2012). <https://doi.org/10.1186/bcr3087> / Zhang H, Wong CC, Wei H, et al. HIF-1-dependent expression of angiopoietin-like 4 and L1CAM mediates vascular metastasis of hypoxic breast cancer cells to the lungs. Oncogene. 2012;31(14):1757–1770 doi:10.1038/onc.2011.365). The authors should reflect on prior literature in the discussion. Perhaps co-injection of shCtrl cells made up for the deficit in HIF1A (by way of growth factor production, etc.) This is at least worth considering as a discussionary point.

6. The overall conclusion of this paper is that breast cancer cells require HIF1A signaling to survive/proliferate in the brain. It is not clear how HIF1A is activated? Do the authors posit that HIF1A is induced in the primary tumor and then somehow maintained even in the bloodstream when the cells are reoxygenated. This would be a very exciting finding but the data as currently presented does not support this conclusion.

7. Do F1 cells metastasize more readily than parental cells. Although proliferation is an important step, many more steps are required for breast cancer cells to home from the primary tumor and colonize the brain niche including overcoming the BBB. Given that proliferation is not a rate limiting step for metastasis, the novelty of the findings would be greatly enhanced if metastatic propensity was considered in addition to proliferation at a metastatic site.

Minor concerns:

1. The present study uses CTCs from 7 patients with hormone receptor-positive metastatic breast cancer. Do the authors think that they would have similar findings if using HER2 and or TNBC CTCs?
2. Supplementary table 1 should show NES and nominal p-value as well.
3. Methods should include a brief description of RNA sequencing experimental setup and library generation.
4. Tables in the supplement have different identifiers. It would be helpful to use the same identifier for cross-referencing purposes?
5. There are many additional references that should be included in the text as this is an emerging field and many labs have reported on the contribution of hypoxia in breast cancer progression and metastasis.

Reviewer #2 (Remarks to the Author):

This manuscript describes an investigation into the molecular factors driving brain metastasis. Using a circulating tumor cell model, the authors generate brain-tropic derivatives and evaluate molecular distinctions from their respective primaries. Hypoxia signaling is identified as upregulated in these derivatives. This observation is then validated in existing transcriptional data and immunofluorescence studies in human samples. Finally, the authors use in vivo approaches to assess the necessity of hypoxia signaling for brain xenograft growth in mice.

This manuscript reports some intriguing observations regarding hypoxia signaling and brain metastasis. However, the reliance of its primary model system on circulating tumor cells is somewhat problematic. Adaptive survival in the circulation likely has significant effects on the molecular rewiring of cancer cells, even exceeding those required for brain colonization. Using CTC-

derived lines as a baseline for transcriptional and functional analyses, therefore, may effectively skew results such that they more reflect the absence of circulation-associated biology than the presence of brain metastasis-associated biology. Additional specific comments are given below.

1) Given the considerations above, perhaps cell lines from mammary xenografts or mammary xenografts themselves (instead of cultured CTC that haven't been passaged through mouse tissue yet) would form a better basis for comparison in these studies. The authors appear to have already generated these reagents for their HIF1A IHC experiments. Transcriptional and functional analyses should be done with these models relative to brain-derived counterparts. These should include interrogation of oxphos and analysis of involved metabolic pathways.

2) For the genetic HIF1A knockdown studies, why wasn't a direct in vivo comparison made in the xenograft studies (flank vs brain, knockdown vs control), as was done for the drug treatment studies? Only a competition assay was performed.

3) the authors report fast and slow growing BM lines from their initial experiments with CTCs (lines 29, 42 vs 82, 50 vs 7, 68, 142). How do these lines compare transcriptionally? Do faster growing lines have expected correlations with HIF1A signaling as do faster growing F1 and F2 derivatives in the isogenic context?

4) All CTC models are HR+, the least likely breast cancers to form brain metastases (vs HER2+ and triple negative variants). This likely impacts the generalizability of findings.

5) FIG. 4F: what are the pathway correlations for this data. Do they resemble those found for the CTC/BM comparisons? What about the mammary/brain xenograft comparisons? Are other metabolic pathways involved (not just hypoxia signaling)?

6) FIG. 3: please document the pathways that are enriched in the F1's. What are the three that overlap between cell types?

7) minor: likely typo in FIG. 4C- "breast" tumors not primary "brain" tumors, correct?

Reviewer #3 (Remarks to the Author):

This paper identifies HIF-1a and associated hypoxia signaling as important pathways for breast cancer metastasis in the brain. This is done by injecting breast CTC derived cell lines into mouse brains and sequencing/ performing metabolic analysis on the parental CTCs and resulting tumors. HIF1a expression was also compared between brain metastasis and breast tumors which showed an increase in HIF1a expression in brain metastasis compared to breast tumors. Furthermore, the paper shows that brain metastasis responded better to dichloroacetic acid (DCA) than breast tumors suggesting that this pathway could be a desirable pathway to target with drugs to better treat brain metastasis.

Hypoxia induced factors are known to play a large role in breast cancer metastasis but less is known regarding their role in brain cancer metastasis. There are only few other studies that look at HIF1a in breast cancer brain metastasis. In vivo Bioluminescence Imaging of Tumor Hypoxia Dynamics of Breast Cancer Brain Metastasis in a Mouse Model (Saha, D. et al. Journal of Visualized Experiments, 2011) transfected MDA MB 231 cells with a HIF1a reporter construct to create a bioluminescence assay before injecting cells into mice brains. They reported an increase in fluorescence which corresponded with the hypoxic environment of the brain. Ebright et al. has expanded on this by performing RNA sequencing and providing further analysis. The other claims of the paper appear to be novel to breast cancer metastasis.

The brain metastasis is associated with high mortality rate and is poorly understood, making this paper of high interest. Additionally, the use of DCA has been tested on some cancers without much use in the field. This paper may help to encourage further exploration into HIF targeted therapies.

However, the following suggestions may strengthen the paper further. In the paper, it is noted that certain CTC cell lines demonstrated better growth after intracranial injection than others and correspond with the brain metastasis in humans. It would be interesting to try to genetically compare CTC lines to see if there are any noteworthy differences that may indicate an increased likelihood of metastasis (ex. Differential gene expression of HIF1a between CTC cell lines). The paper notes that CTCs were authenticated by RNA seq so the data would just need to be reanalyzed. If differences HIF1a expression exist between CTC cell lines this could suggest that HIF targeted therapies could be used even earlier to help combat metastasis. Related to this but outside the immediate scope of this paper, one paper in lung cancer (Wei, DF. et al. Effect of Hypoxia Inducible Factor-1 Alpha on Brain Metastasis from Lung Cancer and Its Mechanism. Sichuan Da Xue Xue Bao Yi Xue Ban. 2019), suggest the increase of HIF1a may result in increase of blood brain barrier permeability.

While the paper does correctly state that the pO₂ of the breast is about 65 mm Hg in breast tissue, in breast tumors the pO₂ ranges from 2.5 to 28 mm Hg, with a median value of 10 mm Hg (Vaupel P, Hockel M, Mayer A. Detection and characterization of tumor hypoxia using pO₂ histography. Antioxid Redox Signal. 2007;9(8):1221–1235.) This would mean that the pO₂ is lower in breast cancer tissue and would not support the claim that breast is less affected by hypoxic signaling. Further clarification on this matter is needed.

Some minor comments

- In figure 1b, BrX-68 is not visible in the in vitro graph
- In figure 2b, axis is labelled relative growt, not growth. Lower error bars also appear to be missing.
- There was no discussion as to why certain in vivo experiments (ex: BrX-29 model in mice brain was ended before other experiments).

Overall, this is an elegant work of high significance and should be accepted after minor revisions.

Sunitha Nagrath

RESPONSE TO REFEREES

Reviewer #1

The manuscript “Hypoxia signaling selectively promotes proliferation of breast cancer metastases in the brain” describes an innovative approach where CTCs from breast cancer patients are harvested from blood samples and maintained in suspension culture in vitro. Cells from these CTCs’ cultures are then implanted in the brain of NSG mice to evaluate their propensity to grow in the brain microenvironment. The authors select 3 of the CTC cell lines with moderate to slow growth potential and enrich these cells by serial intracranial injections. By performing RNA sequencing, the authors conclude that hypoxia signaling is enriched in the derivative cell lines. By performing IHC on matched primary tumor and brain metastases, the authors show a significant increase of HIF1A staining in the brain compared to the lung. Finally, the authors silence HIF1A expression (shRNA), mixed these cells with a scramble control, and demonstrated reduced fraction of shHIF1A cells in brain tumors versus unaffected growth in mammary tumors. There are many publications which the authors should reflect on (by citing) that speak to the role of hypoxia in breast cancer metastasis and the role of HIF1. The novelty of the current study is that serial passage of CTCs in the brain by intracranial injection promotes enhances the ability to proliferate and enriches for a hypoxia signaling program.

We thank the Reviewer for these comments. We apologize for previously neglecting HIF1A-related publications relevant to this work, and we have added these as noted below.

Major concerns:

1. “Hypoxia signaling selectively promotes proliferation of breast cancer metastases in the brain” – title Data does not support title. The study does not examine the role of hypoxia in metastasis. Instead, the experiments show that HIF1A is required for proliferation of breast cancer cells when injected directly into the brain microenvironment. A spontaneous model in which cells exposed to hypoxia in the primary tumor could be “followed” and tested for their ability to spontaneously metastasize would be required to support the title.

We agree with the Reviewer, and we have altered the title of our study to “HIF1A signaling selectively supports proliferation of breast cancer in the brain.”

2. The first experiment is well described and compares the proliferative capacity of 7 distinct patient-derived CTCs in the brain of NSG mice post intracranial injection. The authors highlighted 2 rapid-, 2-moderate and 3-slow proliferating cell lines. They then show images of Ki67 and cleaved Casp3 staining in BRx-82 (moderate-growth cell line and mistakenly indicated as rapid in the text (typo)) versus BRx-142. I am not sure what the goal of this experiment is or how it fits into the context of the rest of the paper which aims to study hypoxic signaling as a mechanism for proficiency to grow in the brain. Do some cell lines have a greater enrichment for the hypoxia signaling than others? Why did the authors choose to continue with 50, 82, and 142 and not the other CTC lines?

We thank the Reviewer for the insightful suggestion to evaluate enrichment for hypoxic signaling in different CTCs lines with variable baseline growth rates in the brain, in addition to the serially passaged isogenic lines. We have now analyzed RNA-seq data from the 7 different CTC lines derived from different patients. To compare relative enrichment for hypoxic signaling, we assessed mean expression of all genes in the Hallmark Hypoxia geneset for each CTC line. Both of the fast-growth lines display significantly higher hypoxic signaling compared with the three slow-growth lines ($p = 0.015$); fast-growth vs moderate-growth lines show a similar trend ($p = 0.070$). In line with these results, the fast-growth lines compared with the slow-growth lines display enrichment for the Hallmark Glycolysis ($p = 0.050$) and Hallmark Angiogenesis ($p = 0.014$) genesets, both of which are downstream targets of hypoxic signaling. These findings further support the observation that hypoxic signaling promotes selective proliferation of breast CTCs in the brain. We have added these data to the manuscript as Figure 3D and updated the text accordingly (page 8, lines 11-22).

The goal of Figure 1 was to identify CTC lines with slow or moderate growth in the brain, so that they could be serially injected to identify mechanisms responsible for increasing and/or promoting proliferation in the brain, especially within the same isogenic background. Two moderate-growth CTC lines (Brx-50, Brx-82) and one slow-growth CTC line (Brx-142) were chosen for further study. This is now clarified in the revised text (page 6, lines 11-16). As noted above, our new data show that intrinsic differences in hypoxic signaling contributes to proliferative differences in the brain confirming the hypoxia correlation that was identified using the serially passaged isogenic lines. We also note that the three CTC lines with the fastest brain growth were derived from women who had intracranial metastases at the time of blood draw, suggesting that the differential properties observed in the mouse model are aligned with clinical history.

3. Please clarify the methods used for the handling conditions of the cells prior to RNA sequencing. If the cells were cultured ex vivo, the experiment suggests that gene expression in the F1 versus parental cell lines that are driven by exposure to the brain niche can be maintained ex vivo? How does this occur? This is an interesting finding and would be supported by the findings of Godet, et al. Nat Commun 10, 4862 (2019). Godet et al shows that non-genomic changes that occur under

hypoxia *in vivo* can be maintained when the cells are returned to tissue culture implying a “hypoxic memory”.

The F1 cells were derived by sorting the dissociated tumor cells for GFP positivity (to eliminate mouse cells) and cultured briefly *ex vivo*, under the same conditions as the parental cells, until sufficient cells were obtained for reinjection and/or experimental analysis. We have clarified this in the revised text (page 6, lines 16-20), as well as in the methods.

We thank the Reviewer for pointing to a recent article showing that tumor cells can maintain hypoxic signaling induced *in vivo* upon subsequent *ex vivo* culture. It is indeed possible that retention of hypoxia-induced signals in CTCs growing within the brain microenvironment contributes to the subsequently enhanced growth of F1 cells in the brain, without affecting their proliferation in the mammary fat pad. We make this point more clearly and now cite the reference above (page 16, lines 11-14).

4. It is unclear whether the CTCs have an “preprogrammed” propensity for increased proliferation in the brain or whether the brain microenvironment reprograms the CTCs. Given that hypoxic signaling is preferentially required for growth in the brain versus the primary site, one could argue that the hypoxia-program only has a benefit in the brain niche. On the other hand, the difference in HIF-1 staining in the brain versus the primary tumor suggests that the hypoxic program is only ‘turned on’ when the cells are localized in the brain. In order to reconcile this question, the authors need to perform sequencing on freshly resected cells sorted from the brain versus the primary tumor. This is critical to determine whether or not hypoxic cells originating in primary tumor have a pre-programmed propensity to survive/proliferate in the brain microenvironment. This would really enhance the novelty of the study.

In the setting of isogenic lines with serial passaging in the brain, the variant levels of hypoxic signaling may be either induced or selected within the primary tumor and then maintained once the tumor cells are cultured. We also note that tumors may be heterogeneous with respect to oxygenation within different parts of the same tumor, adding yet another level of complexity to the question of hypoxic induction vs selection. However, the new data showing that parental fast-growing CTC lines demonstrate elevated hypoxic signaling at baseline without ever having generated a brain tumor in the mouse (response to comment #2) do support the conclusion that this phenomenon, at least in part, reflects intrinsic differences in “brain-competent” CTCs. We feel that these new parental CTC line data (obtained in response to the Reviewer’s comment) address this question more clearly than RNA-seq of primary tumor cells, which could reflect either selection or induction. We now address this important question more clearly in the text (page 16, lines 3-11).

5. To confirm the role of hypoxia signaling on survival and proliferation in the brain, the authors injected a mixture of shHIF1A and shCtrl CTC cells into the brain or into the mammary fat pad. They then show that the fraction of shHIF1a cells was reduced in the brain at the endpoint of the experiment. This was not observed in the mammary tumors. There have been many studies linking primary tumor growth and HIF1A. Please see: Schwab, L.P., Peacock, D.L., Majumdar, D. et al. Hypoxia-inducible factor 1 α promotes primary tumor growth and tumor-initiating cell activity in

breast cancer. *Breast Cancer Res* 14, R6 (2012). <https://doi.org/10.1186/bcr3087> / Zhang H, Wong CC, Wei H, et al. HIF-1-dependent expression of angiopoietin-like 4 and L1CAM mediates vascular metastasis of hypoxic breast cancer cells to the lungs. *Oncogene*. 2012;31(14):1757–1770 doi:10.1038/onc.2011.365). The authors should reflect on prior literature in the discussion. Perhaps co-injection of shCtrl cells made up for the deficit in HIF1A (by way of growth factor production, etc.) This is at least worth considering as a discussionary point.

Given past data on HIF1A expression in cancer, we were also surprised by the minimal impact of HIF1A knockdown on mammary tumor growth in our model. As pointed out by the Reviewer, it is conceivable that part of the previously observed HIF1A-mediated growth advantage in mammary tumors may be non-cell autonomous and rely on the production of growth factors, which would be produced in cell mixing experiments by the control cells (thereby rescuing the HIF1A-KD cells). We now mention this possibility, as well as citing the references listed by the Reviewer. However, we also note that the clear difference in brain vs mammary tumor growth in the cell mixing experiment points to a fundamental difference in these two microenvironments and, moreover, that drug treatment with DCA, a suppressor of glycolysis, preferentially suppresses tumor growth in the brain versus the mammary fat pad. These points are addressed in the revised manuscript (page 17, lines 5-15).

6. The overall conclusion of this paper is that breast cancer cells require HIF1A signaling to survive/proliferate in the brain. It is not clear how HIF1A is activated? Do the authors posit that HIF1A is induced in the primary tumor and then somehow maintained even in the bloodstream when the cells are reoxygenated. This would be a very exciting finding but the data as currently presented does not support this conclusion.

As noted in our response to comment #4, our new data demonstrating increased HIF1A activity in parental CTCs that proliferate rapidly in the brain are most consistent with a model of intrinsic elevated HIF1A signaling within a “brain-competent” subset of CTCs; however, in the context of serial injections of isogenic lines, we cannot differentiate between induction or selection of hypoxic signaling. We have not defined the underlying mechanisms driving the stably increased HIF1A activity in CTCs. That HIF1A transcript levels are not increased in brain-competent CTCs is consistent with post-translational stabilization of HIF1A protein, and while we have not identified VHL mutations in these CTC lines, there are other mechanisms of HIF1A stabilization which may be involved, including tumor-associated hypoxia and “pseudohypoxia”, which comprises a variety of mechanisms leading to reduced HIF1A degradation (1). We now explicitly acknowledge the uncertain mechanism by which HIF1A is stably activated in CTC lines in the text (page 16, lines 3-14).

1. Y. Hayashi *et al.*, Hypoxia/pseudohypoxia-mediated activation of hypoxia-inducible factor-1 α in cancer. *Cancer Sci* **110**, 1510-1517 (2019).

7. Do F1 cells metastasize more readily than parental cells. Although proliferation is an important step, many more steps are required for breast cancer cells to home from the primary tumor and colonize the brain niche including overcoming the BBB. Given that proliferation is not a rate

limiting step for metastasis, the novelty of the findings would be greatly enhanced if metastatic propensity was considered in addition to proliferation at a metastatic site.

We agree with the Reviewer that proliferation is but one step of many in the metastatic cascade, though we note that for disseminated tumor cells, exit from dormancy and proliferation are rate limiting steps in the formation of metastases. As such, a better understanding of the factors that promote growth in the brain microenvironment will guide the development of therapeutic interventions for known breast cancer brain metastases. Moreover, hypoxic signaling pathways attenuate the response to many therapeutic interventions and may contribute to the failure of current treatments in brain metastases.

We also note that, while our F1 lines were generated via serial stereotactic brain injection to study proliferation in the brain microenvironment, we have analyzed publicly available data derived from other models, and we find that hypoxic signaling is enriched in breast cancer cells that have metastasized to the brain following intracardiac injection (hence tissue-specific invasion plus proliferation) (Supplemental Figure 4) (2). Specifically, we analyzed previously published transcriptomic data of two “brain-tropic” cell lines harvested from brain following cardiac injection of parental breast cancer cells. We find that, in addition to the BBB invasion markers reported in the original publication, these brain-tropic daughter cells also have increased hypoxic, angiogenic, and glycolytic signaling, compared to their matched parental cells. Thus, increased hypoxic signaling appears to enhance proliferation of breast cancer cells in the brain, whether they are directly introduced into the brain parenchyma or following the complete blood-based metastatic cascade.

2. P. Bos *et al.*, Genes that mediate breast cancer metastasis to the brain. *Nature* **459**, 1005-1009 (2009).

Minor concerns:

1. The present study uses CTCs from 7 patients with hormone receptor-positive metastatic breast cancer. Do the authors think that they would have similar findings if using HER2 and or TNBC CTCs?

While all of our CTC lines were derived from hormone-receptor positive breast cancers, our reanalysis of data from Bos *et al.*, (2), which used the TNBC cell line MDA-MB-231, demonstrates enrichment of hypoxic signaling in brain-tropic sublines. Nonetheless, there are too few cell lines analyzed in detail within our original manuscript to enable analysis of histology-related effects. However, in the revised text, we have added new clinical data relating to single-cell RNA-seq of primary patient-derived CTCs from women with breast cancer metastatic to brain, in which we find a correlation between reduced overall survival (OS) and increased HIF1A activity. We find that both higher average hypoxic signaling and higher expression of direct HIF1A target genes predict reduced OS following brain metastasis diagnosis (Hypoxia: $p = 0.013$; HIF1A target genes: $p = 0.028$) (Figure 6). These clinical data are comprised of 83 CTCs from 19 patients, including 15 HR+, 2 HER2+ (both HR+/HER2+), and 4 TNBC patients. Covariate analysis demonstrates that the correlation of HIF1A signaling to reduced OS is independent of breast cancer subtype, suggesting a shared

phenomenon across breast cancer subtypes. We discuss these new data in detail in the revised text (page 13, line 18-page 14, line 16).

2. P. Bos *et al.*, Genes that mediate breast cancer metastasis to the brain. *Nature* **459**, 1005-1009 (2009).

2. Supplementary table 1 should show NES and nominal p-value as well.

We have added NES and nominal p-value to Table 1.

3. Methods should include a brief description of RNA sequencing experimental setup and library generation.

We have added RNA-seq library generation and sequencing to the methods.

4. Tables in the supplement have different identifiers. It would be helpful to use the same identifier for cross-referencing purposes?

We apologize for any confusion: there is no overlap between the patients from whom CTC lines were generated (Supplemental Table 1) and the patients from whom matched primary breast and brain metastasis samples were analyzed for HIF1A levels (Supplemental Table 4). Within our newly-added cohort of 19 patients (83 patient-derived CTCs; Figure 6; Supplemental Table 6), there are two cases from whom CTC lines were generated (Brx-42, Brx-82), and these are denoted with the same identifiers. There is no overlap between patients in this new cohort and patients from whom primary breast and brain metastasis samples were analyzed for HIF1A levels. We have clarified these overlaps in the table legends.

5. There are many additional references that should be included in the text as this is an emerging field and many labs have reported on the contribution of hypoxia in breast cancer progression and metastasis.

We apologize for not previously including a more detailed discussion on the contributions of hypoxia to breast cancer progression and metastasis. This is indeed such a large field that we had cited some reviews in addition to key references. We have now made significant additions to the introduction and the discussion, highlighting in greater detail both fundamental roles of hypoxia in tumorigenesis and more novel mechanisms by which hypoxia promotes progression through the metastatic cascade. We have also added 20 references, most of which are related to the role of hypoxia in metastasis.

Reviewer #2

This manuscript describes an investigation into the molecular factors driving brain metastasis. Using a circulating tumor cell model, the authors generate brain-tropic derivatives and evaluate molecular distinctions from their respective primaries. Hypoxia signaling is identified as upregulated in these derivatives. This observation is then validated in existing transcriptional data and immunofluorescence studies in human samples. Finally, the authors use in vivo approaches to assess the necessity of hypoxia signaling for brain xenograft growth in mice.

This manuscript reports some intriguing observations regarding hypoxia signaling and brain metastasis. However, the reliance of its primary model system on circulating tumor cells is somewhat problematic. Adaptive survival in the circulation likely has significant effects on the molecular rewiring of cancer cells, even exceeding those required for brain colonization. Using CTC-derived lines as a baseline for transcriptional and functional analyses, therefore, may effectively skew results such that they more reflect the absence of circulation-associated biology than the presence of brain metastasis-associated biology. Additional specific comments are given below.

We thank the Reviewer for these comments. We agree that cultured CTCs are a unique type of cancer-derived cells which are still poorly characterized, but which have the potential to provide unprecedented patient-derived biology. These cells are cancer cells derived directly from breast cancer patients and captured “in the act” of blood-borne metastasis. As such, differential markers of brain metastasis are particularly interesting. Brain metastases typically occur late in the progression of metastatic breast cancer; hence, it is likely that patient-derived CTCs from such patients are indeed the precursors of brain metastases and they may identify previously unappreciated pathways (versus primary tumor derived models). Moreover, the pronounced differences in brain phenotype by different CTC lines points to their heterogeneity, irrespective of their shared origin from the blood circulation. In the revised manuscript, we now acknowledge more clearly the fact that our observations were derived from patients with advanced breast cancers (page 5, lines 10-14). We also note that we recapitulate our results in reanalysis of previously-published data from the MDA-MD-231 and CN34 cell lines, which are traditional cell lines (page 9, lines 12-20).

1) Given the considerations above, perhaps cell lines from mammary xenografts or mammary xenografts themselves (instead of cultured CTC that haven't been passaged through mouse tissue yet) would form a better basis for comparison in these studies. The authors appear to have already generated these reagents for their HIF1A IHC experiments. Transcriptional and functional analyses should be done with these models relative to brain-derived counterparts. These should include interrogation of oxphos and analysis of involved metabolic pathways.

We thank the Reviewer for this suggestion to conduct comprehensive analyses of xenograft transcriptomic data. As suggested, we have now conducted RNA-seq studies using tissue from mouse brain and mammary gland xenograft tumors. GSEA of tumor-specific transcriptomic data demonstrates upregulation of HIF1A signaling in brain tumors versus mammary tumors, in line with our report of upregulation of hypoxic signaling in brain-competent, isogenic CTC lines. These data are discussed at page 10, lines 11-15. We also

identify several additional pathways that are upregulated in the brain xenograft tumors (Supplemental Table 3).

Importantly, as suggested by the Reviewer in comment #5 and described in detail in response to that comment, we conducted similar transcriptomic analyses of patient primary breast and metastatic brain tumor tissue. In these patient data, we again observe upregulation of HIF1A signaling in brain tumors, as well as additional pathways (Supplemental Table 5; page 11, lines 13-17).

Per the Reviewer's inquiry about additional metabolic pathways beyond hypoxic signaling, we see that SREBP1 signaling is enriched in both mouse and patient brain tumors, suggesting a novel role for lipid/cholesterol metabolism in brain metastasis growth, worthy of future study. This is noted in the revised manuscript (page 11, lines 19-21).

2) For the genetic HIF1A knockdown studies, why wasn't a direct *in vivo* comparison made in the xenograft studies (flank vs brain, knockdown vs control), as was done for the drug treatment studies? Only a competition assay was performed.

As the Reviewer notes, to quantify the effect of HIF1A knockdown, we conducted sequencing-based competition assays of tagged, 1:1-mixed cell cultures, as shown in Figures 5A and 5C, rather than comparing different tumor sizes in different mice. We find this approach to be more quantifiable and reproducible than comparing different mice receiving different tumor cell injections (with inherent mouse to mouse variation and requiring very large numbers of mice for reliable quantitation). Cell mixing experiments are generally considered a reliable and internally controlled approach to measuring cell autonomous proliferation changes, and in addition, the use of sequencing to monitor relative cell growth is more quantitative compared with bioluminescence via *in vivo* imaging, which is inherently noisy. Finally, comparison of two different tissues (brain versus mammary fat pad) is difficult to quantify reliably via *in vivo* imaging, given differential attenuation of luciferase signal passing through different tissue (e.g. greater attenuation of signal passing through skull than through flank skin). In contrast, internally controlled comparisons of brain vs mammary tissue using next-generation sequencing of hairpin tags within each tumor allowed precise quantification of control or knockdown cells. While some noise may

be introduced in this system via preferential PCR amplification, this limitation was mitigated by repeating the experiment with different hairpins.

For the drug treatment studies, such competition assays were impossible, as each mouse received either drug treatment or vehicle, and we therefore had to compare different mice, with either brain or mammary tumors.

3) the authors report fast and slow growing BM lines from their initial experiments with CTCs (lines 29, 42 vs 82, 50 vs 7, 68, 142). How do these lines compare transcriptionally? Do faster growing lines have expected correlations with HIF1A signaling as do faster growing F1 and F2 derivatives in the isogenic context?

We thank the Reviewer for the insightful suggestion to transcriptionally assess the parental CTC lines based on baseline brain growth rate. We have now analyzed RNA-sequencing data from the 7 different CTC lines. To compare relative enrichment for hypoxic signaling, we assessed mean expression of all genes in the Hallmark Hypoxia geneset for each CTC line. Both of the fast-growth lines display significantly higher hypoxic signaling compared with the three slow-growth lines ($p = 0.015$); fast-growth vs moderate-growth lines show a similar trend ($p = 0.070$). In line with these results, the fast-growth lines compared with the slow-growth lines display enrichment for the Hallmark Glycolysis ($p = 0.050$) and Hallmark Angiogenesis ($p = 0.014$) genesets, both of which are downstream targets of hypoxic signaling. These findings are in line with our results from F1 and F2 isogenic lines and further support the observation that hypoxic signaling promotes selective proliferation of breast CTCs in the brain. We have added these data to the manuscript as Figure 3D and updated the text accordingly (page 8, lines 11-22). We believe they are a major contribution to the revised manuscript, and we thank the Reviewer for suggesting these experiments.

4) All CTC models are HR+, the least likely breast cancers to form brain metastases (vs HER2+ and triple negative variants). This likely impacts the generalizability of findings.

While all of our CTC lines were derived from hormone-receptor positive breast cancers, our reanalysis of data from Bos *et al.*, (1), which used the TNBC cell line MDA-MB-231, demonstrates enrichment of hypoxic signaling in brain-tropic sublines. Nonetheless, there

are too few cell lines analyzed in detail within our original manuscript to enable analysis of histology-related effects. However, in the revised text, we have added new clinical data relating to single-cell RNA-seq of primary patient-derived CTCs from women with breast cancer metastatic to brain, in which we find a correlation between reduced overall survival (OS) and increased HIF1A activity. We find that both higher average hypoxic signaling and higher expression of direct HIF1A target genes predict reduced OS following brain metastasis diagnosis (Hypoxia: $p = 0.013$; HIF1A target genes: $p = 0.028$) (Figure 6). These clinical data are comprised of 83 CTCs from 19 patients, including 15 HR+, 2 HER2+ (both HR+/HER2+), and 4 TNBC patients. Covariate analysis demonstrates that the correlation of HIF1A signaling to reduced OS is independent of breast cancer subtype, suggesting a shared phenomenon across breast cancer subtypes. We discuss these new data in detail in the revised text (page 13, line 18-page 14, line 16).

1. P. Bos *et al.*, Genes that mediate breast cancer metastasis to the brain. *Nature* **459**, 1005-1009 (2009).

5) FIG. 4F: what are the pathway correlations for this data. Do they resemble those found for the CTC/BM comparisons? What about the mammary/brain xenograft comparisons? Are other metabolic pathways involved (not just hypoxia signaling)?

To explore this question, we have now conducted RNA-seq on mouse brain and mammary tumors, followed by GSEA for signaling pathways enriched in brain tumors (as described in detail in response to comment #1). We conducted the same gene set enrichment analyses on previously-published patient transcriptomic data of unmatched primary breast tumors and breast cancer brain metastases (2). Genes with HIF1A transcription factor binding sites were significantly enriched in brain tumors versus mammary tumors in both mouse and human, demonstrating functional signaling of the increased HIF1A protein observed in mouse and patient brain tumors, as shown in Figures 4B and 4E and reflecting transcriptomic analysis supporting the data shown in Figures 4C and 4F. These data are

now shown as Figures S7 and S10, and we have updated the text to reflect these additional data (page 10, lines 11-15; page 11, lines 13-17).

Beyond validation of HIF1A transcription factor signaling in brain tumors, these analyses identified several additional pathways enriched in both mouse and patient brain tumors, including SREBP1 signaling, which signals for lipid/cholesterol metabolism. Several reports have suggested that SREBP1 is activated by HIF1A signaling (3, 4); however, we note that there are likely additional factors besides HIF1A signaling that are promoting activation of this pathway in the brain microenvironment. Of note, we did not observe enrichment of SREBP1 signaling in our brain-competent F1 cell lines. Beyond SREBP1, there are several additional pathways enriched in brain tumors. These data are included as Supplemental Tables 3 and 5, and a brief discussion of these additional pathways is included at page 11, lines 19-21.

2. H. Schulten *et al.*, Comprehensive molecular biomarker identification in breast cancer brain metastases. *J Transl Med* **15**, 269 (2017).

3. J. Li *et al.*, Altered metabolic responses to intermittent hypoxia in mice with partial deficiency of hypoxia-inducible factor-1 α . *Physiol Genomics* **25**, 450-457 (2006).

4. E. Furuta *et al.*, Fatty acid synthase gene is up-regulated by hypoxia via activation of Akt and sterol regulatory element binding protein-1. *Cancer Res* **68**, 1003-1011 (2008).

6) FIG. 3: please document the pathways that are enriched in the F1's. What are the three that overlap between cell types?

We thank the Reviewer for pointing out this omission. The three overlapping pathways were Hypoxia (Brx-50 p: <0.001; Brx-82 p: <0.001), KRAS Signaling Up (Brx-50 p: 0.026; Brx-82 p: <0.001), and TNF α Signaling via NF-kB (Brx-50 p: 0.004; Brx-82 FDR: 0.036). We have updated the manuscript with this information (page 7, line 23-page 8, line 3).

7) minor: likely typo in FIG. 4C-“breast” tumors not primary “brain” tumors, correct?

This has been corrected in the revised manuscript.

Reviewer #3

This paper identifies HIF-1a and associated hypoxia signaling as important pathways for breast cancer metastasis in the brain. This is done by injecting breast CTC derived cell lines into mouse brains and sequencing/ performing metabolic analysis on the parental CTCs and resulting tumors. HIF1a expression was also compared between brain metastasis and breast tumors which showed an increase in HIF1a expression in brain metastasis compared to breast tumors. Furthermore, the paper shows that brain metastasis responded better to dichloroacetic acid (DCA) than breast tumors suggesting that this pathway could be a desirable pathway to target with drugs to better treat brain metastasis.

Hypoxia induced factors are known to play a large role in breast cancer metastasis but less is known regarding their role in brain cancer metastasis. There are only few other studies that look at HIF1a in breast cancer brain metastasis. In vivo Bioluminescence Imaging of Tumor Hypoxia Dynamics of Breast Cancer Brain Metastasis in a Mouse Model (Saha, D. et al. Journal of Visualized Experiments, 2011) transfected MDA MB 231 cells with a HIF1a reporter construct to create a bioluminescence assay before injecting cells into mice brains. They reported an increase in fluorescence which corresponded with the hypoxic environment of the brain. Ebright et al. has expanded on this by performing RNA sequencing and providing further analysis. The other claims of the paper appear to be novel to breast cancer metastasis.

The brain metastasis is associated with high mortality rate and is poorly understood, making this paper of high interest. Additionally, the use of DCA has been tested on some cancers without much use in the field. This paper may help to encourage further exploration into HIF targeted therapies.

(1) However, the following suggestions may strengthen the paper further. In the paper, it is noted that certain CTC cell lines demonstrated better growth after intracranial injection than others and correspond with the brain metastasis in humans. It would be interesting to try to genetically compare CTC lines to see if there are any noteworthy differences that may indicate an increased likelihood of metastasis (ex. Differential gene expression of HIF1a between CTC cell lines). The paper notes that CTCs were authenticated by RNA seq so the data would just need to be reanalyzed. If differences HIF1a expression exist between CTC cell lines this could suggest that HIF targeted therapies could be used even earlier to help combat metastasis.

We thank the Reviewer for the insightful suggestion to transcriptionally assess the parental CTC lines based on baseline brain growth rate. We have now analyzed RNA-sequencing data from the 7 different CTC lines. To compare relative enrichment for hypoxic signaling, we assessed mean expression of all genes in the Hallmark Hypoxia geneset for each CTC line. Both of the fast-growth lines display significantly higher hypoxic signaling compared with the three slow-growth lines ($p = 0.015$); fast-growth vs moderate-growth lines show a similar trend ($p = 0.070$). In line with these results, the fast-growth lines compared with the slow-growth lines display enrichment for the Hallmark Glycolysis ($p = 0.050$) and Hallmark Angiogenesis ($p = 0.014$) genesets, both of which are downstream targets of hypoxic signaling. These findings are in line with our results from F1 and F2 isogenic lines and

further support the observation that hypoxic signaling promotes selective proliferation of breast CTCs in the brain. We have added these data to the manuscript as Figure 3D and updated the text accordingly (page 8, lines 11-22). We believe they are a major contribution to the revised manuscript, and we thank the Reviewer for suggesting these experiments.

With respect to HIF1A mRNA expression, the two fast-growth CTC lines have the highest HIF1A RPM; however, the difference in HIF1A RPM between fast-growth lines and the other lines is minimal and not significant (as shown below). This is consistent with known post-translational regulation of HIF1A, and differences in expression of downstream HIF1A targets serve as a reflection of increased HIF1A protein and activity.

	Brx-7	Brx-29	Brx-42	Brx-50	Brx-68	Brx-82	Brx-142
	slow	fast	fast	moderate	slow	moderate	slow
HIF1A	122.51	154.46	199.66	98.18	134.12	118.53	151.16

(2) Related to this but outside the immediate scope of this paper, one paper in lung cancer (Wei, DF. et al. Effect of Hypoxia Inducible Factor-1 Alpha on Brain Metastasis from Lung Cancer and Its Mechanism. Sichuan Da Xue Xue Bao Yi Xue Ban. 2019), suggest the increase of HIF1a may result in increase of blood brain barrier permeability.

We thank the Reviewer for this interesting article, which highlights that HIF1A may promote crossing of the BBB, in addition to promoting growth in the brain microenvironment, as indicated by our data. We have included this reference in the discussion (page 15, lines 7-10).

(3) While the paper does correctly state that the pO₂ of the breast is about 65 mm Hg in breast tissue, in breast tumors the pO₂ ranges from 2.5 to 28 mm Hg, with a median value of 10 mm Hg (Vaupel P, Hockel M, Mayer A. Detection and characterization of tumor hypoxia using pO₂ histography. Antioxid Redox Signal. 2007;9(8):1221–1235.) This would mean that the pO₂ is lower in breast cancer tissue and would not support the claim that breast is less affected by hypoxic signaling. Further clarification on this matter is needed.

We thank the Reviewer for bringing forth this important point. Per the highlighted meta-analysis, the median pO₂ of breast tumors is 10 mmHg, though the vast majority of values < 10mmHg come from just 2 of the 10 studies analyzed in the meta-analysis, suggesting potential downward skewing of the data. Unfortunately, there is little published on the pO₂ levels of breast cancer brain metastases; however, in one cohort, median pO₂ was 4.4 mmHg (1). Additionally, areas experiencing hypoxia in brain metastases are larger than areas experiencing hypoxia in primary breast tumors, suggesting that breast cancer brain metastases face both more profound and more widespread hypoxia versus primary breast tumors. We have added a discussion in the text to highlight these data as potential contributors to the increased HIF1A levels and signaling we observe in brain metastases; however, we note that there may be other or additional factors promoting HIF1A signaling, as well (page 16, line 16-page 17, line 3).

Additionally, though in a CNS-derived cancer, we note that multiple studies have demonstrated median pO₂ levels of ~5 mmHg in glioblastoma, with even more widespread hypoxic areas than in breast cancer brain metastases, further suggesting the relevance of low oxygenation on tumors growing in the brain microenvironment (2, 3).

1. P. Vaupel *et al.*, Detection and characterization of tumor hypoxia using pO₂ hisography. *Antioxid Redox Signal* **9**, 1221-1235 (2007).
2. R. Rampling *et al.*, Direct Measurement of pO₂ distribution and bioreductive enzymes in human malignant brain tumors. *Int J Radiat Oncol Biol Phys* **29**, 427-431 (1994).
3. D. R. Collingridge *et al.*, Polarographic measurements of oxygen tension in human glioma and surrounding peritumoural brain tissue. *Radiother oncol* **53**, 127-131 (1999).

Some minor comments

(1) In figure 1b, BrX-68 is not visible in the *in vitro* graph

This was due to Brx-68 and Brx-7 having nearly-perfectly overlapping *in vitro* growth curves. To address this issue, we have changed the symbols for these samples to allow both to be seen.

(2) In figure 2b, axis is labelled relative growt, not growth. Lower error bars also appear to be missing.

The typo in the Figure 2B axis has been corrected.

With respect to the lower error bars in Figure 2B, we had initially not shown them, since the log₁₀ y-axis results in asymmetric lower and upper error bars, which can be confusing to the reader. We have now incorporated the lower error bars and adjusted the figure.

(3) There was no discussion as to why certain *in vivo* experiments (ex: BrX-29 model in mice brain was ended before other experiments).

The Brx-29 and Brx-42 arms of the brain growth experiment described in Figure 1B were ended early, as per IACUC protocol, since the brain tumors grew to the point of causing distress to the animals. We now state this explicitly in the text (page 5, lines 18-21).

(4) Overall, this is an elegant work of high significance and should be accepted after minor revisions.

We thank the Reviewer for the positive comments and for her thoughtful and constructive review.

REVIEWERS' COMMENTS

Reviewer #1 (Remarks to the Author):

Thank your for the additions that were made during the revision process. They clarified the minor questions that we had. I have no further concerns.

Reviewer #2 (Remarks to the Author):

I have no further issues with this manuscript. The authors should be commended for their diligence in addressing reviewer comments.

Reviewer #3 (Remarks to the Author):

The revised manuscript sufficiently addressed the concerns previously raised. It can be accepted for the publication without any further changes.

RESPONSE TO REFEREES

Reviewer #1

Thank your for the additions that were made during the revision process. They clarified the minor questions that we had. I have no further concerns.

Reviewer #2

I have no further issues with this manuscript. The authors should be commended for their diligence in addressing reviewer comments.

Reviewer #3

The revised manuscript sufficiently addressed the concerns previously raised. It can be accepted for the publication without any further changes.

We thank all of the Reviewers for the insightful and helpful comments, which significantly elevated the importance, comprehensiveness, and clarity of our findings.